# The carbon cycle in the Australian Community Climate and Earth System Simulator (ACCESS-ESM1). 2. Historical simulations

Tilo Ziehn[1], Andrew Lenton[2], Rachel M. Law[1], Richard J. Matear[2], and Matthew A. Chamberlain[2]

[1]CSIRO Oceans and Atmosphere, PMB 1, Aspendale, Victoria, Australia
[2]CSIRO Oceans and Atmosphere, Hobart, Tasmania, Australia

*Correspondence to:* Tilo Ziehn (tilo.ziehn@csiro.au)

**Abstract.** Over the last decade many climate models have evolved into earth system models (ESMs), which are able to simulate both physical and biogeochemical processes through the inclusion of additional components such as the carbon cycle. The Australian Community Climate and Earth System Simulator (ACCESS) has been recently extended to include land and ocean carbon cycle components in its ACCESS-ESM1 version. A detailed description of ACCESS-ESM1 components including results from pre-industrial simulations is provided in Part 1. Here, we focus on the evaluation of ACCESS-ESM1 over the historical period (1850-2005) in terms of its capability to reproduce climate and carbon related variables. Comparisons are performed with observations, if available, but also with other ESMs to highlight common weaknesses. We find that climate variables controlling the exchange of carbon are well reproduced. However, the aerosol forcing in ACCESS-ESM1 is somewhat larger than in other models, which leads to an overly strong cooling response in the land from about 1960 onwards. The land carbon cycle is evaluated for two scenarios: running with a prescribed leaf area index (LAI) and running with a prognostic LAI. We overestimate the seasonal mean (1.7 vs. 1.4) and peak amplitude (2.0 vs. 1.8) of the prognostic LAI at the global scale, which is common amongst CMIP5 ESMs. However, the prognostic LAI is our preferred choice, because it allows for the vegetation feedback through the coupling between LAI and the leaf carbon pool. Our globally integrated land-atmosphere flux over the historical period is $98\,\mathrm{PgC}$ for prescribed LAI and $137\,\mathrm{PgC}$ for prognostic LAI, which is in line with estimates of land-use emissions (ACCESS-ESM1 does not include land-use change). The integrated ocean-atmosphere flux is $83\,\mathrm{PgC}$, which is in agreement with a recent estimate of $82\,\mathrm{PgC}$ from the Global Carbon Project for the period 1959 to 2005. The seasonal cycle of simulated atmospheric $CO_2$ is close to the observed seasonal cycle (up to $1\,\mathrm{ppm}$ difference for station at Mace Head and up to $2\,\mathrm{ppm}$ for station at Mauna Loa), but shows a larger amplitude (up to $6\,\mathrm{ppm}$) in the high northern latitudes. Overall, ACCESS-ESM1 performs well over the historical period, making it a useful tool to explore the change in land and oceanic carbon uptake in the future.

## 1 Introduction

Climate models are continuously evolving to include more processes and interactions at higher resolutions and their number has increased rapidly in recent years. In addition, a number of institutes worldwide have been developing earth system models

(ESMs), which are able to simulate both physical and biogeochemical processes through the inclusion of the land and ocean carbon cycles.

The evaluation of ESMs in terms of their capability to reproduce climate and carbon related variables over the historical period (i.e. 1850 to 2005) is crucial prior to using such models for future predictions. Comparisons are usually performed with observation based products, if available, but also with other ESMs to identify common weaknesses.

The performance of 18 ESMs that participated in the Coupled Model Intercomparison Project phase 5 (CMIP5) (Taylor et al., 2012) has been evaluated in Anav et al. (2013) for the present day climate. They found that all models correctly reproduce the main climate variables controlling the spatial and temporal variability of the carbon cycle. However, large differences exist when reproducing specific fields. In terms of the land carbon cycle, an overestimation of photosynthesis and leaf area index (LAI) was found for most of the models. In contrast, for the ocean an underestimation of the net primary production (NPP) was noted for a number of models. Anav et al. (2013) also found significant regional variations in model performance.

Eight of these CMIP5 ESMs were also evaluated in Shao et al. (2013), highlighting that temporal correlations between annual-mean carbon cycle and climate variables vary substantially among the 8 models. Large inter-model disagreements were found for NPP and heterotrophic respiration (Rh). In agreement with Anav et al. (2013), Shao et al. (2013) also noted that the CMIP5 historical simulations tend to overestimate photosynthesis and LAI.

Todd-Brown et al. (2013) compared and evaluated 11 CMIP5 ESMs in terms of their variations in soil carbon. The correct representation of soil carbon in the model is important in order to accurately predict future climate-carbon feedbacks. Soil carbon simulations of the 11 models were compared against empirical data from the Harmonized World Soil Database (HWSD) and from the Northern Circumpolar Soil Carbon Database (NCSCD). A large spread across all models was found (nearly 6 fold) and the spatial distribution of soil carbon, especially in the northern latitudes was found to be poor in comparison to HWSD and NCSCD, which means that most ESMs were poorly representing grid-scale soil carbon.

Frölicher et al. (2015) showed that CMIP5 models appeared to capture the observed pattern of anthropogenic carbon storage in the ocean, particularly in the Southern Ocean. However, overall they underestimate the magnitude of the observed oceanic global anthropogenic carbon storage since the pre-industrial.

The representation of the global carbon cycle in ESMs continues to be challenging. For example, large uncertainties exist for the climate-carbon feedback, which can be mainly attributed to terrestrial carbon cycle components (Friedlingstein et al., 2006; Arora et al., 2013). Terrestrial ecosystem models show large variations when driven with future climate scenarios (Shao et al., 2013; Friend et al., 2014) due to differences in model formulation and uncertainties in process parameters (Knorr and Heimann, 2001; Booth et al., 2012).

The Australian Community Climate and Earth System Simulator (ACCESS) participated in CMIP5, but in a climate model only version. A selection of CMIP5 simulations have now been performed with the ESM version of ACCESS, ACCESS-ESM1 (Law et al., 2015). Here, we present the performance of the land and ocean carbon cycle components of ACCESS-ESM1 over the historical period (1850-2005). First, we briefly assess ACCESS-ESM1 simulation of climate variables that are relevant to the carbon cycle (Sect. 3). We then focus on the response of the carbon cycle to the historical forcing (Sect. 4) and comparison

of various present-day simulated carbon variables with observations (Sect. 5). Law et al. (2015) provides complementary analysis of the ACCESS-ESM1 pre-industrial simulation.

## 2    Model configuration, simulations and comparison data

Historical simulations (Sect. 2.2) are performed with two model configurations (Sect. 2.1) and the results compared with other CMIP5 ESMs (Sect. 2.3) and a number of observed data products (Sect. 2.4).

### 2.1    Model configuration

ACCESS-ESM1 is based on the ACCESS climate model (Bi et al., 2013), but with the addition of biogeochemical components for ocean and land as described in part 1 of this paper (Law et al., 2015). The climate model version underlying the ESM version is ACCESS1.4, a minor update of the ACCESS1.3 version submitted to CMIP5 (Bi et al., 2013; Dix et al., 2013). The relationship between the ACCESS1.3, ACCESS1.4 and ACCESS-ESM1 versions is illustrated in Law et al. (2015, Fig. 1). Law et al. (2015) also showed that the climate simulations of the three model versions are very similar.

For the ACCESS-ESM1 version, ocean carbon fluxes are simulated by the World Ocean Model of Biogeochemistry And Trophic dynamics (WOMBAT) (Oke et al., 2013) and land carbon fluxes are simulated by the Community Atmosphere Biosphere Land Exchange (CABLE) model (Kowalczyk et al., 2006; Wang et al., 2011), which optionally includes nutrient limitation (nitrogen and phosphorus) for the terrestrial biosphere through its biogeochemical module, denoted CASA-CNP (Wang et al., 2010). This capability is important because nitrogen, phosphorus and carbon biogeochemical cycles are strongly coupled, and it has been demonstrated that nutrient limitation has a large impact on the productivity of terrestrial ecosystems (Wang et al., 2010; Goll et al., 2012; Zhang et al., 2013). Consequently, global land carbon uptake can be altered significantly. Here we run CASA-CNP in 'CNP' mode with both nitrogen and phosphorus limitation active. This differentiates the ACCESS-ESM1 simulations presented here from other ESM simulations for CMIP5, few of which included nitrogen and none of which included phosphorus.

As in Law et al. (2015), two model configurations are used, differing in their treatment of leaf area index (LAI). LAI is an important variable in climate models for describing the biophysical and biogeochemical properties of the land cover and in CABLE it can either be prescribed or simulated. When prescribed, monthly values based on MODIS observations are read in through an external file (Law et al., 2015, Sec. 3.1.1). The dataset used here is limited by having no interannual or longer time-scale variability. Additionally the same LAI is assigned to all plant funtional types (PFTs) within a grid-cell even though CABLE simulates multiple PFTs per grid-cell. With prescribed LAI there is no coupling between the LAI and the leaf carbon pool which means that vegetation feedbacks cannot be included. These limitations are removed by making LAI a prognostic variable with the LAI dependent on the simulated size of the leaf carbon pool. However if the leaf carbon pool is not well simulated then this would lead to a poor LAI simulation with consequent impacts for the climate simulation.

## 2.2 Simulations

All experiments are set up as concentration driven simulations, which means that (historical) atmospheric $CO_2$ concentrations are prescribed as an input to ACCESS-ESM1 and changes in the land and ocean carbon pools do not feed back on to atmospheric $CO_2$ concentrations following CMIP5 protocols (Taylor et al., 2012).

As noted above we run ACCESS-ESM1 in two configurations, with prescribed LAI (PresLAI) and prognostic LAI (ProgLAI). For PresLAI, the carbon cycle has no impact on the simulated climate whereas for ProgLAI, there is a small impact on the climate through biogeophysical feedbacks related to surface albedo, evaporation and transpiration (Law et al., 2015, Sec. 4.1). The difference in LAI will also have an impact on the land carbon fluxes, whereas the impact on the ocean carbon cycle is negligible, and therefore our analysis of the ocean carbon fluxes focuses only on one scenario (i.e. PresLAI).

Both configurations of ACCESS-ESM1 were run for 1000 years under pre-industrial climate conditions (year 1850) (Law et al., 2015) with the historical simulations starting from year 800 of these control runs. As noted in Law et al. (2015) the net carbon fluxes for land and ocean did not equilibrate to zero. At the end of the control run (i.e. year 800 to 955), global NEE is $0.3 \, \mathrm{PgC \, yr^{-1}}$ for PresLAI and $0.08 \, \mathrm{PgC \, yr^{-1}}$ for ProgLAI. The net autgassing from the ocean is about $0.6 \, \mathrm{PgC \, yr^{-1}}$ at the end of the control run. We take this drift into account when we calculate the net uptake of carbon for land and ocean.

The historical simulations use external forcing for 1850-2005 such as increasing greenhouse gases, aerosols, changes in solar radiation and volcanic eruptions as used in previous ACCESS versions (Dix et al., 2013). For example, the prescribed atmospheric $CO_2$ increases from 285 ppm in 1850 to 379 ppm in 2005.

     Volcanic eruptions in ACCESS-ESM1 are prescribed based on monthly global mean stratospheric volcanic aerosol optical depth (Sato et al., 2002) which is then averaged over four equal-area latitude zones, similar to the way it is done in the

20 Hadley Centre Global Environmental Model (HadGEM) (Stott et al., 2006; Jones et al., 2011). Globally significant volcanoes within the historical period are Krakatoa (1883), Santa Maria (1903), Agung (1963), El Chichón (1982) and Pinatubo (1991). Tropospheric aerosols are either calculated interactively (i.e. sea salt and mineral dust) or are based on emission datasets (i.e. sulphate and organic carbon) and increase rapidly from 1950 (Dix et al., 2013, Fig. 4).

     The simulations do not include any land-use change; the distribution of PFTs used in the pre-industrial simulation is used

throughout the historical period.

## 2.3 Comparison with CMIP5 models

ACCESS-ESM1 is compared against other ESMs that participated in CMIP5 and are available on the Earth System Grid. The models used in this paper are shown in Table 1 with the references provided in Lenton et al. (2015). As not all years were available for these simulations, we focused on the period 1870-2005 and used only the first ensemble member for each ESM.

In assessing the response of the CMIP5 models, we calculated the median and the 10th and 90th percentiles following Lenton et al. (2015). This allows us to both assess how well ACCESS-ESM1 captures the median and whether it falls into the range of existing CMIP5 models.

## 2.4 Observations

We use the following observational data products to compare against ACCESS-ESM1 outputs. Climate variables are assessed, where this is helpful for interpreting the carbon simulation. For example, the land carbon balance is mainly controlled by surface temperature and precipitation (Piao et al., 2009), whereas the ocean carbon balance is mainly influenced by sea surface temperature (SST) and mixed layer depth (MLD) (Martinez et al., 2009).

*Land surface temperature and precipitation:* Climate Research Unit (CRU) 1901-2013 time-series (TS) data set at version 3.22 (Harris et al., 2014; Jones and Harris, 2014), statistically interpolated to 0.5° x 0.5° from monthly observations at meteorological stations across the world's land area (excluding Antarctica). A low resolution version at 5° for land surface temperature anomalies (CRUTEM4, (Jones et al., 2012)) is used for the period 1850-1900.

*Sea surface temperatures (SST):* the high-resolution (1° x 1°) Hadley SST1 (Rayner et al., 2003) in the period 1870-2006. We also use data from the World Ocean Atlas climatology (WOA2005; Garcia et al., 2006a, b) in the Taylor diagram.

*Climatological mixed layer depths:* de Boyer Montégut et al. (2004) for the historical period, based on the density mixed layer criteria of a change density of $0.03 \, \mathrm{kg \, m^{-3}}$ from the surface.

*Ocean net primary productivity (NPP):* from SeaWIFS calculated with the VPGM algorithm of Behrenfeld and Falkowski (1997).

*Global ocean and land carbon flux:* Global Carbon Project (GCP) estimates of annual global carbon budget components and their uncertainties using a combination of data, algorithms, statistics and model estimates (Le Quéré et al., 2015). The GCP residual land sink is estimated as the difference of emissions from fossil fuel and cement production, emissions from land use and land cover change (LULCC), atmospheric $CO_2$ growth rate and the mean ocean $CO_2$ sink. The 2014 global carbon budget (Le Quéré et al., 2015) provides annual values for the period 1959 to 2013.

*Gross primary production (GPP):* upscaled data from the Flux Network (FLUXNET) using eddy covariance flux data and various diagnostic models (Beer et al., 2010). Gridded data at the global scale is provided by Jung et al. (2011) using a machine learning technique called model tree ensemble (MTE) to scale up FLUXNET observations. Global flux fields are available at a 0.5° x 0.5° spatial resolution and a monthly temporal resolution from 1982 to 2008.

*LAI:* global LAI derived from the third generation (3g) Global Inventory Modeling and Mapping Studies (GIMMS) normalized difference vegetation index (NDVI)3g data set. Neural networks were trained first with best-quality and significantly postprocessed Moderate Resolution Imaging Spectroradiometer (MODIS) LAI and Very High Resolution Radiometer (AVHRR) GIMMS NDVI3g data for the overlapping period (2000 to 2009) to derive the final data set at 1/12° resolution and a temporal resolution of 15 days for the period 1981 to 2011 (Zhu et al., 2013).

*Soil organic carbon (SOC):* the Harmonized World Soil Database (HWSD) (FAO, 2012) represents the most comprehensive and detailed globally consistent database of soil characteristics that is currently available for global analysis. We use an upscaled and regridded version of the HWSD with the area weighted SOC calculated from the soil organic carbon (%), bulk density and soil depth (Wieder et al., 2014).

*Phosphate, salinity, DIC and alkalinity:* observations for phosphate and salinity come from the World Ocean Atlas climatology (WOA2005; Garcia et al., 2006a, b), while DIC and alkalinity are from GLODAP (Key et al., 2004).

*Sea-air* $CO_2$ *fluxes:* seasonal climatology of Wanninkhof et al. (2013) based on the $1°$ x $1°$ global measurements of oceanic $pCO_2$ of Takahashi et al. (2009).

*Anthropogenic carbon uptake:* column inventory estimated from Sabine et al. (2004) from GLobal Ocean Data Analysis Project (GLODAP) (Key et al., 2004).

*Atmospheric* $CO_2$ *concentrations:* mean atmospheric $CO_2$ seasonal cycles derived from NOAA/ESRL flask samples as processed in the GLOBALVIEW (GLOBALVIEW-$CO_2$, 2011) data product. These seasonal cycles are designed to be representative of background, clean-air at any given location. Here, we assess the seasonal cycle for 4 locations with an averaging period of about 20 years for Mace Head ($53.33°$ N, $9.90°$ W), about 25 years for Alert ($82.45°$ N, $62.52°$ W), about 35 years for South Pole ($89.98°$ S, $24.80°$ W) and about 40 years for Mauna Loa ($19.53°$ N, $155.58°$ W).

## 2.5 Performance evaluation

For climate variables such as land surface temperature and precipitation we calculate the model variability index (MVI) (Gleckler et al., 2008; Scherrer, 2011). The models ($mod$) variability at every grid point $i$ is compared against the observed ($obs$) variability and then averaged over the globe in the following way:

$$\text{MVI} = \frac{1}{n} \sum_{i=1}^{n} \left( \frac{s_i^{mod}}{s_i^{obs}} - \frac{s_i^{obs}}{s_i^{mod}} \right)^2, \tag{1}$$

where $s$ is the standard deviation and $n$ the number of grid cells. Perfect model - observations agreement would result in an MVI of zero. The definition of a limit to decide if a model performs well or poor is rather arbitrary. However, Scherrer (2011) and Anav et al. (2013) have used a threshold of MVI$< 0.5$.

For a number of carbon related variables we calculate the inter-annual variability (IAV), defined as the standard deviation of detrended annual mean values.

To assess the performance of the ocean carbon cycle against observations we use a Taylor diagram (Taylor, 2001). We also apply the same analysis to archived CMIP5 simulations (Taylor et al., 2012) to benchmark the performance of ACCESS-ESM1 relative to other CMIP5 models. A Taylor diagram allows us to summarise the bias, relative variability and correlations of the simulations with the observations. In the plot, the radial distance of a given simulation from the origin gives the standard deviation of the simulation normalised by the standard deviation of the observations. The angle from the $x$ axis provides the spatial correlation coefficient between the simulations and the observations. The radial distance from the point marked observations gives a measure of the RMS difference between the simulation and observations normalised by the standard deviation of the observations. The point's colour represent the bias in the simulation given as the relative difference in the globally averaged values between simulation and observations calculated as (mean_model – mean_observations)/mean_observations; positive values show the model is overestimating the observed value.

## 3 ACCESS-ESM1 climatology

### 3.1 Land temperature and precipitation

Carbon fluxes across the historical period will be directly influenced by increasing atmospheric $CO_2$ and indirectly influenced by changes in the climate, driven by the increasing atmospheric $CO_2$ and modulated by other external forcings, such as anthropogenic and volcanic aerosols. In addition, each climate simulation generates its own internal variability, with major modes of climate variability such as the El Niño Southern Oscillation (ENSO) known to generate large variability in carbon exchange between the atmosphere and both the ocean and land (Zeng et al., 2005).

The evolution of temperature and precipitation in ACCESS-ESM1 (Fig. 1) over land shows similar characteristics to ACCESS1.3 historical simulations (Dix et al., 2013; Lewis and Karoly, 2014) as well as those of ACCESS1.4 (P. Vohlarik, pers. comm.). Global land surface air temperature anomalies (relative to 1901-1930) are shown in Fig. 1. Both ACCESS-ESM1 simulation scenarios (PresLAI and ProgLAI) show similar temperature anomalies over most of the historical period, being close to the observed anomalies through most of the period (decadal mean difference smaller than 0.2 K), apart from the 1940s where the PresLAI scenario shows a larger negative anomaly (decadal mean difference of about 0.37 K), which will be discussed later. From about 1965-2005 anomalies are by up to 0.4 K (decadal mean difference) lower than observations for both scenarios. This is attributed by Lewis and Karoly (2014) to a likely overly strong cooling response in ACCESS1.3 to anthropogenic aerosols, offsetting the warming due to greenhouse gas increases for which ACCESS1.3 responds similarly to a CMIP5 mean (Lewis and Karoly, 2014, Figs. 2a, 3a). Strong aerosol cooling is supported by Rotstayn et al. (2015) who found that ACCESS1.3 showed a large global mean aerosol effective radiative forcing (ERF) over the historical period of $-1.56\,\mathrm{W\,m^{-2}}$ which is much larger than the IPCC best estimate ($-0.9\,\mathrm{W\,m^{-2}}$) (Boucher et al., 2013) but still within the uncertainty range.

The interannual variability in temperature is well reproduced by both ACCESS-ESM1 scenarios, showing an MVI of 0.3 (PresLAI) and 0.4 (ProgLAI) for the period 1901-2005. According to Anav et al. (2013) only a few CMIP5 models show an MVI of lower than 0.5 (although their calculation is based on present day, i.e. 1986-2005).

Both ACCESS-ESM1 simulations exhibit cooling following major volcanic eruptions (marked in Fig. 1). At first sight, the ProgLAI run seems to be more sensitive to volcanic eruptions, showing a stronger cooling particularly for the two most recent major eruptions, El Chichón in 1982 and Mt. Pinatubo in 1991. However, this difference might be due to a different ENSO phase for the two runs at the time of the eruptions. Lewis and Karoly (2014) assessed the temperature impact of Agung, El Chichón and Pinatubo in three ACCESS1.3 simulations (e.g. their Fig. 7) and mean temperature anomalies from the two ACCESS-ESM1 simulations lie within or only slightly outside the ACCESS1.3 ensemble range. It is worth noting that Lewis and Karoly (2014) found that the simulated temperature anomalies from volcanoes tended to be larger in ACCESS than observed, and this was common across CMIP5 models.

Differences in the year to year temperature anomalies between the two ACCESS-ESM1 scenarios are likely due to internal climate variability. For example, between the years 1940 and 1950, the PresLAI run shows a large negative temperature anomaly and the ProgLAI run shows a positive anomaly. The negative anomaly for the PresLAI is probably related to a strong

La Niña event (Nino3 index of -1.2) around the year 1945 (Fig. 1c), whereas in the ProgLAI case we see a small El Niño event (Nino3 index of 0.6) around the same time.

The temperature anomalies hide an absolute temperature difference between the two ACCESS-ESM1 simulations; the ProgLAI scenario produces a slightly warmer climate (0.56 K difference in mean land surface air temperature averaged over 1850-2005) than the PresLAI run. This is consistent with the difference in surface air temperature found for the pre-industrial simulations (Law et al., 2015, Sec. 4.1). As noted in Law et al. (2015) the warmer climate can be explained by the difference in LAI, which is generally higher in the prognostic case. This leads to a lower albedo, especially for evergreen needleleaf forests during the winter months in the northern hemisphere, and consequently to an increase in absorbed radiation. The difference in LAI for both scenarios is explored in more detail in section 5.1.2. Compared to the observations the ACCESS-ESM1 runs show a cooler land surface air temperature by about 0.5 K for the ProgLAI scenario and 1.1 K for the PresLAI scenario averaged over 1901-2005.

Precipitation anomalies over the land are presented in Fig. 1b. Larger differences in the anomalies for the two ACCESS-ESM1 simulations can be observed around the years 1870 to 1880, where the PresLAI scenario shows a positive anomaly and the ProgLAI scenario shows a mainly negative anomaly. The difference over the remaining time period for the two runs is generally small. ACCESS-ESM1 simulations compare well with observed rainfall anomalies until about 1960 (decadal mean difference smaller than $8\,\mathrm{mm\,yr^{-1}}$), with the exeption of the period 1911-1920 for PresLAI (decadal mean difference of about $12\,\mathrm{mm\,yr^{-1}}$) and the period 1951-1960 for ProgLAI (decadal mean difference of about $17\,\mathrm{mm\,yr^{-1}}$). After that, observed anomalies are mostly higher than the simulation results (decadal mean difference of up to $41\,\mathrm{mm\,yr^{-1}}$), a feature also seen in the ACCESS1.3 historical ensemble (Lewis and Karoly, 2014, Fig. 6a). The comparison of absolute rainfall for the two ACCESS-ESM1 scenarios suggests a dryer climate (approx. $20\,\mathrm{mm\,yr^{-1}}$) for the ProgLAI run.

For precipitation we calculate an MVI of 1.7 (PresLAI) and 1.8 (ProgLAI) for the period 1901-2005, which suggests that the IAV is not well represented in ACCESS-ESM1. However, according to Anav et al. (2013) none of the CMIP5 models had an MVI close to the threshold of 0.5. Also note that for the calculation of the MVI for precipitation we had to exclude 60 land points (mainly coastal points) due to inconsistancies in the regridding.

A reduction in precipitation can be observed following the eruption of major volcanoes for both ACCESS-ESM1 scenarios, apart from the 1903 Santa Maria eruption and the 1982 El Chichón eruption, where the PresLAI scenario does not show a strong anomaly and the ProgLAI anomaly is likely too late to be due to the volcano. As for temperature, the precipitation anomalies lie within or close to the ACCESS1.3 ensemble of anomalies presented by Lewis and Karoly (2014, Fig. 9).

## 3.2 Sea surface temperature and mixed layer depth

To assist in the assessment of responses of the ocean NPP and sea-air $CO_2$ fluxes, the responses of SST and mixed layer depth are first assessed.

The ocean response from ACCESS-ESM1 is compared with the time series of HadiSST v1 (Rayner et al., 2003) in Figure 2. Here we see, that there is a warm bias in the early part of the historical period. This warm bias in ACCESS-ESM1 is the same as reported by Bi et al. (2013) over the period 1870-1899 in ACCESS 1.3 (0.26 K). In the period 1870-1970 we see that

the warming of the oceans appears to be less climate sensitive than the observations. However, by the end of the historical simulation (1970-2005) we notice that ACCESS-ESM1 captures well the observed response of HadiSST in the later period.

However, despite little global bias in the latter period we see that the ACCESS-ESM1 SST response, consistent with AC-CESS 1.3 (Bi et al., 2013), produces strong spatial differences from observations. Fig. 3 shows clear spatially coherent differences between ACCESS-ESM1 and observations (1986-2005). Some of these regions show a strong summer warming bias (>3 K) in areas such as the high latitude Southern and Pacific Ocean, while in other regions such as the subtropical Atlantic, a strong cooling bias is present during the same season. This is in contrast to other regions, such as the high latitude North Atlantic, that has a strong year round warming bias. These biases are broadly consistent with known errors associated with the UK Met Office Unified Model (Williams et al., 2015), which is employed as the atmospheric model in ACCESS-ESM1. Our SST response is also broadly consistent with other ESMs such as HadGEM2 (Martin et al., 2011) that also use the UK Met Office Unified Model.

The magnitude of the interannual variability of simulated SST is of similar magnitude as the observations. In response to large aerosol injections associated with volcanic eruptions, overlain on Fig. 2, we see that the ocean does capture a net cooling, as expected (e.g. Stenchikov et al., 2009) and consistent with observations. Interestingly, the magnitude of the cooling is sometimes less than observed in HadiSST v1 despite the stronger than observed aerosol response in ACCESS-ESM1.

Ocean mixed layer depths are compared with the observations following de Boyer Montégut et al. (2004), based on more than $880000$ depth profiles from research ships and ARGO profiles, and based on a $0.03 \, \mathrm{kg \, m^{-3}}$ density change from the surface. Significant advances in autonomous measurement platforms have allowed the mixed layer to be increasingly well constrained in all seasons across the global ocean.

Overall we see in the mid and lower latitudes that the mixed layer depth is deeper than observed in all seasons (Figure 4). However the very large values likely represent the differences in the positions of fronts between the relatively coarse resolution model relative to the observations rather than very large differences (Lenton et al., 2013). In the higher latitudes winter mixed layers are well captured by ACCESS-ESM1 (Figure 4). This is encouraging given that many ocean models tend to underestimate winter mixed layer depths (Sallée et al., 2013; Downes et al., 2015). Simulating winter mixed layers correctly is critical for setting interior ocean properties supplying nutrients to the upper ocean to fuel the biologically active growing season (Rodgers et al., 2014). However in contrast to the winter, ACCESS-ESM1 appears to systematically underestimate mixed layer depths in the high latitude ocean in summer, $60\%$ (or $30\text{-}40 \, \mathrm{m}$) in the Southern Ocean, Pacific and Atlantic Oceans. In the Southern Ocean, in particular, the underestimation of summer mixed layer depths is consistent with Sallée et al. (2013) and Huang et al. (2014) who showed that most CMIP5 models underestimate summer mixed layer depths. Huang et al. (2014) attributed this to a lack of vertical mixing in CMIP5 rather than sea surface forcing related to individual models, this is consistent with Downes et al. (2015), who showed that these biases are also present in the ocean only simulations of ACCESS-ESM1.

## 4  ACCESS-ESM1 carbon cycle response to historical forcing

The increase in atmospheric $CO_2$ over the historical period is expected to have a direct impact on both land and ocean carbon fluxes. Additionally there may be indirect impacts from the change in climate caused by the increasing atmospheric $CO_2$. These impacts are explored firstly for land carbon and then for ocean carbon.

### 4.1  Land carbon response

The direct impact of increasing atmospheric $CO_2$ is seen clearly in the simulated global land gross primary production (GPP) (Fig. 5a), with increasing GPP for both simulations. The ProgLAI case gives the larger increase, with fluxes for the final 10 years of the simulation being 19% larger than for the first 10 years, compared to an increase of 11% in the PresLAI case. This is due to increasing LAI in the ProgLAI simulation (Fig. 5b) compared to the prescribed LAI which is annually repeating with no increase. Thus the PresLAI case captures only the direct $CO_2$ fertilisation effect of more efficient photosynthesis per leaf area while the ProgLAI case also allows the growing leaf biomass to increase the global total assimilation. The inter-annual variability (IAV) in GPP over the whole historical period for the ProgLAI run is $2.6\,\mathrm{PgC\,yr^{-1}}$, considerably larger than in the PresLAI case ($1.7\,\mathrm{PgC\,yr^{-1}}$), but within the range of other CMIP5 models. We also notice a large decadal variability of global GPP for the ProgLAI case, which is much weaker in the PresLAI case (1.9 vs. 1.3 $\mathrm{PgC\,yr^{-1}}$ ). Natural variability of the climate is the main driver for the IAV in GPP for the PresLAI case. The larger variability in the ProgLAI case is due to the stronger response to volcanic cooling and climate, causing an increase in LAI and a positive feedback through increased GPP. In the PresLAI case, without the LAI feedback, the impact of volcanic cooling is sometimes largely offset by natural climate variability, for example in the Pinatubo (1991) case.

The difference between the two simulations is less obvious for the net ecosystem exchange (Fig. 5c). NEE is a relatively small flux that represents the difference between respiration (heterotrophic and autotrophic) and GPP. In the current set up of ACCESS-ESM1 we do not include disturbances such as fire and LULCC, which means that in this case NEE also represents the net flux of carbon from the land to the atmosphere. Both simulations generally produce small land sinks over most of the historical period, with some tendency to an increasing sink from the 1920s, followed by a possible reduction in the sink from the mid 1990s to 2005. The IAV is relatively large and similar for both scenarios (1.4 vs. 1.3 $\mathrm{PgC\,yr^{-1}}$) and likely caused by variations in GPP (Piao et al., 2009; Jung et al., 2011) that are moderated by respiration, especially in the ProgLAI case. Law et al. (2015, Table 2) found similar IAV in the preindustrial simulation with larger GPP IAV in the ProgLAI case offset by positively correlated leaf respiration IAV. Decadal variability for the ProgLAI run is larger than for the PresLAI run (0.7 vs. 0.3 $\mathrm{PgC\,yr^{-1}}$).

Larger decadal variability in the ProgLAI run can be explained by the stronger response to volcanic eruptions. In principle, aerosols scatter incoming solar radiation and therefore have a mainly cooling effect. Hence, an increase in aerosol emissions leads to a decrease in global temperature which in turn increases GPP in the tropics and reduces plant respiration globally in both cases (PresLAI and ProgLAI) and therefore increases NEE. However, whereas in the PresLAI case the LAI is kept at a

constant level, in the ProgLAI case the LAI is allowed to increase with the leaf carbon pools (Fig. 5b). This leads to a further increase in GPP at the same time (Fig. 5a) which further increases NEE in the ProgLAI case.

Due to the fact that during the control run our net carbon flux did not equilibrate to zero (Law et al., 2015, Sec. 4.2.2), we calculate the carbon uptake for both scenarios by subtracting the mean net flux over the corresponding part of the control run. We estimate a total uptake of carbon to the land (using the net ecosystem production (NEP), with $NEP = -1 \times NEE$) over the historical period of 98 PgC for the PresLAI scenario and 137 PgC for the ProgLAI scenario. The increase in biomass over the historical period is 70 PgC for PresLAI and 87 PgC for ProgLAI, (see also Table 2). This is similar to results from CMIP5 models that also do not consider LULCC. For, example the Beijing Climate Center Climate System Model (BCC-CSM1.1) estimates an increase in biomass of about 83 PgC over the historical period and the Institute of Numerical Mathematics Coupled Model (INM-CM4.0) reports an increase of about 70 PgC (Jones et al., 2013). The increase in combined soil and litter carbon over the historical period is smaller in ACCESS-ESM1 (28 PgC for PresLAI and 49 PgC for ProgLAI) than in the two CMIP5 models without LULCC (64 PgC for both, BCC-CSM1.1 and INM-CM4.0).

We can compare the total carbon uptake (here cumulative NEP) from ACCESS-ESM1 with other models and estimates in two ways:

1. Comparison against land-use emission estimates:

   The observation based cumulative historical land carbon uptake is estimated to be $-11 \pm 47$ PgC (Arora et al., 2011), which suggests an almost neutral behaviour of the land over that period. Since we do not include disturbances in our model, we do not expect our simulations to match those results. However, we can compare our calculated cumulative uptake against estimates of land-use emissions to see if they are in a similar range. For example, Houghton (2010) reports land-use emissions of 108–188 PgC for 1850-2000, comparable to the ACCESS-ESM1 cumulative uptakes.

2. Comparison against CMIP5 estimates of cumulative NEP:

   Simulation results from CMIP5 ESMs that include LULCC provide a large range for the total carbon uptake. Shao et al. (2013, Table 4), for example, reports the separate contributions of NEP and disturbance to cumulative land carbon uptake for eight CMIP5 models. While NEP ranges from 24-1730 (median 387) PgC and disturbance ranges from 3-1729 PgC, the range for land uptake is smaller with two outlying models (-120 and 211 PgC) and the remainder ranging from -59 to 18 PgC. The estimates of cumulative NEP from ACCESS-ESM1 are at the low end of the CMIP5 range reported in Shao et al. (2013), possibly due to the inclusion of nitrogen (N) and phosphorus (P) limitation; Zhang et al. (2013) found a reduction of 1850-2005 NEP from 210 PgC for a carbon-only simulation to 85 PgC with N and P limitation when using CABLE in a low resolution earth system model.

## 4.2 Ocean carbon response

Figure 6 shows that, consistent with other CMIP5 models, there is no statistically significant trend of ocean NPP globally over the historical period. The global mean NPP from ACCESS-ESM1 of 51 PgC yr$^{-1}$ is close to that calculated from the SeaWIFS data of 50 PgC yr$^{-1}$ for 1998-2005. Furthermore it is also in agreement with estimates, based on observations, of global NPP

of between 45-50 $PgC yr^{-1}$ (Behrenfeld and Falkowski, 1997). The ACCESS-ESM1 NPP is larger than the median CMIP5 model value of 37 PgC, however NPP in CMIP5 models is associated with a very large range (Anav et al., 2013).

The evolution of sea-air $CO_2$ fluxes in the period 1850-2005 is shown in Fig. 7. Overlain on this plot is the timing of the major volcanic eruptions, the estimated sea-air $CO_2$ flux from the Global Carbon Project (GCP) (Le Quéré et al., 2015) and
5 results from the CMIP5 model archive. We also take into account the drift over the corresponding part of the control run. Here we see very good agreement with the CMIP5 models in the period 1870-1960, with the ACCESS-ESM1 sitting close to the median of the CMIP5 models, and well within the range of the CMIP5 models. After 1960, ACCESS-ESM1 shows greater uptake than the median of CMIP5 models, and appears to more closely follow the observed value from the GCP, lying at the 10th percentile of the CMIP5 range. For 1960-2005, ACCESS-ESM1 gives a mean sea-air $CO_2$ flux of $1.8 \pm 0.1 \, PgC yr^{-1}$
in good agreement with the estimated GCP value of $1.9 \pm 0.3 \, PgC yr^{-1}$, and larger than the estimate from CMIP5 models of $1.56 \pm 0.1 \, PgC yr^{-1}$. For 1986-2005, the sea-air $CO_2$ is $2.2 \pm 0.1 \, PgC yr^{-1}$ from ACCESS-ESM1, the same as from the GCP ($2.2 \pm 0.2 \, PgC yr^{-1}$), and larger than the median CMIP5 model value of $1.8 \pm 0.1 \, PgC yr^{-1}$. The cumulative uptake of carbon by air-sea $CO_2$ fluxes in the period 1959-2005 from ACCESS-ESM1 is 83 PgC which is good agreement with the GCP value of 82 PgC (Le Quéré et al., 2015) over the same period. These results highlight that ACCESS-ESM1 show good skill at
capturing the globally integrated ocean carbon uptake at the global scale.

## 5   Evaluation of the present day carbon cycle

The last 20 years of the historical simulation (1986-2005) is used to evaluate the simulated carbon cycle against observation based products. Analysis considers the land, ocean and atmosphere in turn.

### 5.1   Land carbon

### 5.1.1   GPP

Both ACCESS-ESM1 runs (PresLAI and ProgLAI) provide a mean GPP of about 130 $PgC yr^{-1}$ for 1986-2005. The observation based estimate of Jung et al. (2011) suggests a GPP of about 119 $PgC yr^{-1}$ for the same period. Other studies also suggest a global GPP within the same range: Beer et al. (2010) reports an estimate also based on FLUXNET data of $123 \pm 8 \, PgC yr^{-1}$ for the period 1998-2005; Ziehn et al. (2011) used plant traits to constrain parameters of the Farquhar photosynthesis model
and estimated the global GPP for the same period to be 121 $PgC yr^{-1}$ (95% confidence interval from 110 to 130 $PgC yr^{-1}$) and the IPCC in its AR4 report states a global value of 120 PgC for 1995 (Denman et al., 2007). If compared with other CMIP5 earth system models which were divided into two groups by Anav et al. (2013), ACCESS-ESM1 lies in the middle of the lower group with the range 106 to 140 $PgC yr^{-1}$. It was also noted by Anav et al. (2013), that the group of CMIP5 models with a GPP above 150 PgC did not include nitrogen limitation and might therefore overestimate GPP. ACCESS-ESM1 contains
both nitrogen and phosphorus limitation, which may provide a more realistic simulation of carbon uptake by the terrestrial biosphere.

A number of studies that base their estimates on observations suggest that a global GPP of about $120\,\mathrm{PgC\,yr^{-1}}$ may be somewhat too low. For example, Welp et al. (2011) provides a best guess of $150\text{-}175\,\mathrm{PgC\,yr^{-1}}$ and (Koffi et al., 2012) an estimate of $146 \pm 19\,\mathrm{PgC\,yr^{-1}}$. However, the estimate by Jung et al. (2011) is based on the largest set of observations and also provides a spatial distribution of GPP. In the following, we therefore use this product for the validation of the ACCESS-ESM1 land carbon component.

The mean annual cycle of GPP as simulated by the ACCESS-ESM1 is shown in Fig. 8 for both scenarios as Anav et al. (2013, Fig. 8). Observation based estimates by Jung et al. (2011) are also shown for comparison. At the global scale both ACCESS-ESM1 runs show a similar behaviour and they both overestimate GPP by about $2\,\mathrm{PgC\,month^{-1}}$ (peak amplitude) if compared with the observations as discussed earlier. However, when we split GPP into its contributions from three latitudinal regions we notice larger differences between the two ACCESS-ESM1 simulations. The ProgLAI simulation shows a much more productive northern region (by about $2\,\mathrm{PgC\,month^{-1}}$) and a lower GPP in the tropics (by about $0.2\,\mathrm{PgC\,month^{-1}}$), which compensated for at the global scale. Overall, both ACCESS-ESM1 simulations show good agreement with the observations in terms of the amplitude, with only a small bias of up to $2.2\,\mathrm{PgC\,month^{-1}}$ for the globe and the northern hemisphere. In contrast, a large number of CMIP5 models produce a strong positive bias during June-August on a global scale and for the northern hemisphere (Anav et al., 2013). Agreement with observations in terms of the phase is generally good, accept for the Tropics, where ACCESS-ESM1 fails to accurately reproduce the phase. However, as noted by Anav et al. (2013) this is common amongst CMIP5 models.

The spatial distribution of GPP is presented in Fig. 9 along with its IAV for the last 20 years of the historical period. Generally there is good agreement in the spatial pattern of GPP between ACCESS-ESM1 with prescribed LAI and the observation based estimate ($95\%$ of all land points have errors smaller than $0.5\,\mathrm{kgC\,m^{-2}\,yr^{-1}}$). However, there are some small differences mainly in tropical regions (i.e. central Africa). The ACCESS-ESM1 ProgLAI run shows a larger GPP in the NH, mostly in the boreal regions, and a lower GPP for large parts of South-America ($86\%$ of all land points have errors smaller than $0.5\,\mathrm{kgC\,m^{-2}\,yr^{-1}}$). Comparing the IAV of GPP for the two ACCESS-ESM1 runs reveals large differences. Whereas the PresLAI run shows little variability for most areas, the ProgLAI run shows large hotspots in South-America and Southeast Australia of up to $0.5\,\mathrm{kgC\,m^{-2}\,yr^{-1}}$ which are caused by the LAI feedback as discussed previously. The observation based estimate of GPP shows large areas of variability over the continents, but the distribution and magnitude are quite different to the ACCESS-ESM1 runs. However, as pointed out in Anav et al. (2013) one of the limitations of the GPP observational product is the magnitude of the IAV.

### 5.1.2 LAI

Global LAI estimates are mainly derived from satellite observations and various products are available. The prescribed LAI in ACCESS-ESM1 is based on MODIS observations (Yang et al., 2006) with no IAV. If compared with the observation based estimates of Zhu et al. (2013), which uses a combination of MODIS and AVHRR data, over the last 20 years of the historical period (mean of 1.4), we notice that our current prescribed LAI is somewhat smaller (mean of 1.3), but agrees well in terms of

its seasonal cycle (Fig. 10). There is a number of reasons why remote sensing LAI products differ from each other, i.e. because different sensors and algorithms are used (Los et al., 2000).

The prognostic LAI which is calculated by CASA-CNP is significantly higher at the global scale (mean: 1.7) and also shows a different seasonality with its peak in August, whereas the observations suggest the peak is in July (Fig. 10). In CABLE the phenology phase is currently prescribed and the leaf onset might be defined as too late for deciduous vegetation which leads to a shift in the LAI peak by about one month.

The global seasonal cycle of LAI is mainly influenced by the northern extra-tropics and we notice that leaf coverage through-out the year and especially in autumn and winter is too high in the ProgLAI case. We clearly overestimate the mean LAI (observations suggest a mean of 1.3) and underestimate the seasonal variability. On a PFT level the main contributor to this is evergreen needle leaf forest which produces a large value (mean 3.8) over the whole year with only a very small seasonal cycle. In the tropics we underestimate LAI by a significant amount (mean of 1.5 in comparison to 2.3 as suggested by observations). This is mainly due to C4 grass showing an LAI which is about a factor of 5 smaller than the observations. Law et al. (2015) attributes the low simulated LAI of C4 grass to a large sensitivity to rainfall and the inability of CABLE to grow back C4 grass after a die back.

The overestimation of the LAI for evergreen needle leaf forest and the underestimation for C4 grass have a direct impact on GPP, which is also too large for evergreen needle leaf and too low for C4 grass. In CABLE, the calculation of GPP is related to APAR (absorbed photosynthetic active radiation) which is the product of FPAR (fraction of photosynthetically active radiation) and PAR (photosynthetically active radiation) with FPAR calculated from the LAI.

At the global scale, most CMIP5 earth system models also tend to overestimate LAI (Anav et al., 2013, Fig. 11), ranging from 1.5 in December-January to almost 3.5 in June-August. Anav et al. (2013) reports that only 2 models captured the main feature of the global LAI pattern, whereas the remaining 16 models overestimate the global LAI with some even exceeding a mean of 2.4. At the regional scale the ACCESS-ESM1 prognostic LAI is within the CMIP5 range for both hemispheres, but below the CMIP5 range for the Tropics.

### 5.1.3 NEE

We compare our NEE results against estimates of the residual land sink from the global carbon project (GCP) (Le Quéré et al., 2015) for 1959–2005 (Fig. 5c). The mean residual land sink and interannual variability for this period is estimated to be about $1.9\pm1.0\,\mathrm{PgC\,y^{-1}}$ compared to $1.4\pm1.3\,\mathrm{PgC\,y^{-1}}$ for PresLAI and $1.8\pm1.6\,\mathrm{PgC\,y^{-1}}$ for ProgLAI. In all cases the IAV is large relative to the mean uptake, but more so in the ACCESS-ESM1 simulations. The large IAV makes it difficult to be definitive about land uptake trends over this period, though there is some suggestion of slightly increasing uptake in the GCP budget estimates but slightly decreasing uptake in the ACCESS-ESM1 simulations. This might be better assessed using an ensemble of simulations and extending the analysis closer to 2015 through use of the RCP scenario simulations. Simulations without anthropogenic aerosols would also be useful to determine whether the relatively strong cooling due to tropospheric aerosols in ACCESS-ESM1 is impacting the decadal evolution of land carbon uptake.

### 5.1.4 CNP pool sizes

The amount of carbon, nitrogen and phosphorus stored in the biomass and soil of terrestrial ecosystems as simulated by ACCESS-ESM1 is compared against other estimates from the literature. Here, we refer to the terrestrial biomass as the sum of living above ground (leaf and wood) and below ground (roots) material. All mean pool sizes and spatial distributions derived from ACCESS-ESM1 are calculated over the last 20 years of the historical period (1986-2005).

Carbon pool sizes simulated with ACCESS-ESM1 are in general smaller for the PresLAI scenario as shown in Table 2. The total carbon in the terrestrial biomass amounts to 670 PgC (PresLAI) and 807 PgC (ProgLAI). The IPCC (Prentice et al., 2001) reports two different estimates of 466 PgC and 654 PgC for the global plant carbon stock, depending on the data being used. This would imply that our plant carbon pools are somewhat to large, especially for the ProgLAI scenario. However, we have to take into account account that we do not consider LULCC, which might be the reason why we overestimate the size of our carbon pools. Other studies such as Houghton et al. (2009) suggest a range of 800-1300 PgC for the global terrestrial biomass. The large range is a result of inconsistent definitions of forest, uncertain estimates of forest area, paucity of ground measurements and the lack of reliable mechanisms for upscaling ground measurements to larger areas (Houghton et al., 2009).

A large number of observational based estimates for global soil organic carbon (SOC) exists with most studies reporting a global estimate of about 1500 PgC (Scharlemann et al., 2014). SOC pools simulated by ACCESS-ESM1 are somewhat smaller with 1050 PgC for the PresLAI scenario and about 1200 PgC for the ProgLAI scenario. However, these numbers agree well with the best estimate of 1260 PgC derived from the HWSD (FAO, 2012) and considering the large range of 510 - 3040 PgC of global SOC simulated by CMIP5 models (Todd-Brown et al., 2013) this is an encouraging result.

The Harmonized World Soil Database (HWSD) also provides a spatial distribution of the SOC density which is shown in Fig.11 along with the results from ACCESS-ESM1. In general there is good agreement between the two ACCESS-ESM1 scenarios, showing a similar pattern, but with a slightly larger density in the NH boreal region for the ProgLAI run. The agreement between the HWSD and ACCESS-ESM1 is also generally good. However, the HWSD suggest localized hot spots of high SOC density in North America and Siberia which are not covered by ACCESS-ESM1. We also underestimate SOC in the tropics especially in the maritime continent region. On the other hand, both ACCESS-ESM1 scenarios suggest a high SOC density in the north Asian region, which is not apparent in the HWSD.

In addition to other environmental constraints such as water, light and temperature, carbon storage by terrestrial ecosystems may also be limited by nutrients, predominantly nitrogen and phosphorus (Wang and Houlton, 2009; Wang et al., 2010; Zhang et al., 2013). However, few estimates are available of total nitrogen and phosphorus pool sizes and their global spatial distribution is even more uncertain.

Simulated nitrogen pool sizes are shown in Table 2, and there is only a small difference between the two ACCESS-ESM1 scenarios. Our estimate for the nitrogen in the terrestrial biomass is about 6.5 PgN. Estimates based on field data reconstructions range from about 3.5 PgN (Schlesinger, 1997) to 10 PgN (Davidson, 1994) which places the ACCESS-ESM1 results right in the middle of that range. Soil organic nitrogen pools are simulated to be about 85 PgN for both ACCESS-ESM1 scenarios which is slightly low if compared with estimates based on field data (95 PgC (Post et al., 1985) to 140 PgC (Batjes, 1996)).

The terrestrial phosphorus cycle at present day is even less constrained than the nitrogen cycle and modelling and empirical estimates vary greatly. ACCESS-ESM1 results suggest a total of 0.35 PgP in the terrestrial biosphere which is lower than the estimated range of 0.5 - 1 PgP by Smil (2000). Organic soil phosphorus pool sizes differ to some extent between the two ACCESS-ESM1 scenarios. The PresLAI model run simulates a pool size of about 10 PgP and the ProgLAI model run gives a pool size of about 12 PgP (see Table 2). Other estimates range from about 5 PgP to about 200 PgP with the upper end being assessed as unrealistic (Smil, 2000).

## 5.2 Ocean carbon

### 5.2.1 Surface field assessment

Figure 12 shows the Taylor diagram comparing the mean surface phosphate, oxygen, alkalinity, DIC, temperature and salinity fields. The ACCESS-ESM1 surface fields are 20-year averages (1986-2005), assessed against observations. Overlain on this plot are median values from CMIP5. The individual CMIP5 models are listed in Table 1.

For all variables considered, ACCESS-ESM1 simulations show correlations with the observations of better than 0.6. SST shows the highest correlation ($R > 0.98$) with observations having a better correlation and lower bias than the median of the CMIP5 models, and a very similar magnitude of variability with the observations. In contrast, sea surface salinity (SSS) shows the lowest correlation in both ACCESS-ESM1 and CMIP5 median. ACCESS-ESM1 underestimates the observed SSS variability and had a global mean value that is less than observed. The median of the CMIP5 models has a slightly lower correlation with the observations but with greater variability and greater underestimate of the global averaged SSS than ACCESS-ESM1. Biases in SSS are not surprising given the challenges with capturing well the hydrological cycle in ESMs (Trenberth et al., 2003).

The poor representation of salinity in ACCESS-ESM1 and the CMIP5 simulations will impact the simulated alkalinity. ACCESS-ESM1 has large regional biases in surface salinity (Bi et al., 2013, Fig. 16), which produce surface alkalinity biases because alkalinity is strongly influenced by air–sea freshwater exchanges. The ACCESS-ESM1 has a similar correlation with the observations as SSS but with a weak negative bias. To reduce the alkalinity bias of ACCESS-ESM1 one needs to reduce the export of calcium carbonate from the upper ocean. However, given the strong relationship between SSS and surface alkalinity the higher correlation of alkalinity in the CMIP5 median with the observations suggests these models may be over-tuning their simulation to compensate for the errors in the SSS simulation.

For DIC, ACCESS-ESM1 shows a good correlation with observations (Fig. 12), which is comparable with CMIP5 median, but overestimates the magnitude of the variability when compared with CMIP5 and observations. The underestimation of the mean value, also seen in CMIP5 median, can be attributed to the negative alkalinity bias reducing the surface DIC concentration that would be in equilibrium with the atmosphere. In comparison to the observations and CMIP5, phosphate is poorly represented in ACCESS-ESM1 with a large overestimation of the surface concentration. In contrast, the median from CMIP5 shows relatively small bias to the observed value. Despite a poor representation of phosphate, this does not translate into a significant bias in global primary productivity.

While assessing the simulated values with the median CMIP5 values provides valuable insight, it does not allow us to assess the skill of our model with individual CMIP5 models. To do this the simulated surface DIC and alkalinity values are compared with individual CMIP5 models (Fig. 13). For alkalinity (Fig. 13a), ACCESS-ESM1 shows a similar correlation as the CMIP5 models. The CMIP5 models range between under and over-estimating the surface alkalinity concentration. While the ACCESS-ESM1 has a negative bias in surface alkalinity it is still within the range of the CMIP5 models. For DIC, we see that our simulation sits in the middle of the CMIP5 correlation values with the lowest RMS error with the observations (Fig. 13b). All simulations show negative DIC biases and the ACCESS-ESM1 is not a significant outlier. Overall, our simulation has comparable skill to the existing CMIP5 models.

### 5.2.2 Net primary production

To assess the seasonal anomaly of ocean NPP, calculated as the anomaly of vertically integrated primary productivity through the water column, the global ocean is broken down into 5 regions, following (Anav et al., 2013). Figure 14 shows the NPP seasonal anomaly from ACCESS-ESM1, CMIP5 models and SeaWIFS over the (SeaWIFS) observational period 1998-2005. At the global ocean scale, seasonally we see that the magnitude of NPP from ACCESS-ESM1 is less than the amplitude of CMIP5 and SeaWIFS, with poor phasing. This likely reflects the biases in ACCESS-ESM1 toward lower latitudes, reflecting excess nutrient supply, and utilization, to the upper oligotrophic ocean (Law et al., 2015) associated with deeper than observed mixed layers. In the northern and southern subtropical gyres ACCESS-ESM1 (18 N-49 N and 19 S-44 S respectively) appears to overestimate the amplitude of the observed seasonal cycle when compared with SeaWIFS. Again this overestimate of NPP is associated with deeper than observed mixed layers which increase nutrient supply to the oligotrophic upper ocean. The phase of the NPP in these regions, where agreement between observations and CMIP5 is very good, is delayed by about three months. This delay may also be explained by a combination of higher (than observed) concentrations of nutrients and slower than expected biological productions associated with cool biases, particularly in the Atlantic Ocean allowing the bloom to occur later.

In the high latitude northern hemisphere, the magnitude of the seasonal cycle of NPP is not well captured in ACCESS-ESM1. While CMIP5 appears also to underestimate the magnitude of the seasonal cycle, ACCESS-ESM1 is lower again. In contrast, in the Southern Ocean the amplitude of the seasonal cycle of NPP in ACCESS-ESM1 shows good agreement with observations. However in the high latitude oceans the phase of NPP is delayed by about 2 months. This delay may be attributed to the too shallow mixed layers that exist in these regions, which means that it is only when mixed layers start to deepen that biological productivity can start to occur. As a result the remaining growing season is shorter (than observed) leading to a reduced total productivity. This may in part explain why the total NPP northern hemisphere is much less than observed.

Interestingly, in the tropical ocean we see very good agreement in the amplitude of the seasonal cycle with CMIP5 and SeaWIFS. We note however, that comparing the phase of the seasonal cycle from ESMs (ACCESS-ESM1 and CMIP5) with SeaWIFS is not very meaningful in this region, as they all simulate their own ENSO cycle with their own timing. Therefore, any comparison over a 20 year period between models has the potential to be biased by the number of El Niño or La Niña events.

### 5.2.3 Sea-air CO$_2$ fluxes

Figure 15 shows that, in the period 1986-2005, ACCESS-ESM1 is in good agreement with the spatial pattern and the magnitude of sea-air CO$_2$ fluxes of Wanninkhof et al. (2013), hereafter referred to as W13. In the Southern Ocean (44 S-90 S), which is an important net sink of carbon, ACCESS-ESM1 (-0.77 PgC yr$^{-1}$) captures a larger annual mean uptake than the sea-air CO$_2$ flux of W13 who only estimated an uptake of -0.18 PgC yr$^{-1}$. In the Southern subtropical gyres (44 S-18 S) ACCESS-ESM1 (-0.39 PgC yr$^{-1}$) captures, but overestimates, the observed sea-air flux of W13 (-0.23 PgC yr$^{-1}$). In contrast in the Northern Hemisphere ACCESS-ESM1 underestimates the uptake at -0.36 PgC yr$^{-1}$ and -0.19 PgC yr$^{-1}$ in the subtropical, and (sub) polar regions respectively, while W13 estimated the uptake at -0.69 PgC yr$^{-1}$ and -0.54 PgC yr$^{-1}$ over the same regions. The uptake in the tropical ocean is well captured, showing very good agreement between ACESS-ESM1 and W13 who estimate an uptake of -0.56 PgC yr$^{-1}$ and -0.57 PgC yr$^{-1}$. Spatially the interannual variability in sea-air CO$_2$ flux is presented in a companion paper (Law et al., 2015).

The anomaly of the seasonal cycle of the sea-air CO$_2$ fluxes was assessed against observations of W13 and CMIP5, shown in Fig.16 for the period 1986-2005. Here, we see that ACCESS-ESM1 has a larger global amplitude of sea-air CO$_2$ fluxes than observed (W13) and simulated, but close to the upper value of the range from CMIP5 models. We also see that globally the phase of sea-air CO$_2$ fluxes is not well captured in ACCESS-ESM1, lying outside the range of the CMIP5 models. To better understand why there are differences between ACCESS-ESM1, CMIP5 and W13 we separate the response of sea-air CO$_2$ into the same regions as for NPP, again following Anav et al. (2013).

ACCESS-ESM1 appears to capture well the phase of sea-air CO$_2$ fluxes in the subtropical gyres. In the northern subtropical gyre in particular, we see that the amplitude and phase of the seasonal cycle in ACCESS-ESM1 shows very good agreement with W13, in contrast with other ESMs (CMIP5). In the southern subtropical gyres, while the ACCESS-ESM1 appears to overestimate the amplitude relative to the observations, we see very good agreement with CMIP5 models. As anticipated the tropical ocean shows very little seasonality, nevertheless we do see good agreement with CMIP5 models. However, the comparison of ACCESS-ESM1 against observations (while shown) is not very meaningful as W13 is based on values of oceanic pCO$_2$ from Takahashi et al. (2009), which does not include El Niño years.

The largest differences are seen in the representation of sea-air CO$_2$ fluxes in the high latitude ocean. In the high latitude northern hemisphere, we see that the magnitude is larger than either CMIP5 or W13 and shows poor phasing. While the magnitude of the seasonal cycle in the Southern Ocean lies within the upper range of CMIP5 again poor phasing is seen. That the seasonal cycle is out of phase suggests that during the summer the solubility response likely dominates over the NPP response, leading to an out-gassing in the summer and uptake in the winter, as discussed in Lenton et al. (2013). Consequently, we see that the poor global phasing in global sea-air CO$_2$ fluxes is likely due to the solubility dominated response of the high latitudes during the summer.

### 5.2.4 Anthropogenic inventory

The global inventory of anthropogenic carbon from ACCESS-ESM1 is compared with the uptake from GLODAP (Sabine et al., 2004) for the year 1994 in Fig. 17. Here we see that the spatial pattern of the column inventory of anthropogenic carbon is very well reproduced, with the large storage occurring in the North Atlantic and large uptake in the Southern Ocean. The inventory for the period 1850–1994 in ACCESS-ESM1 is 132 PgC, which is close to the estimated value from GLODAP of $118\pm19$ PgC (Sabine et al., 2004) over the same domain. This suggests that despite a somewhat limited representation of the seasonal cycle of sea-air $CO_2$ fluxes in key regions of anthropogenic uptake such as the Southern Ocean, that ACCESS-ESM1 is doing a very good job, spatially and temporally, of capturing and storing anthropogenic carbon. If the entire domain (including the Arctic Ocean) the is integrated the anthropogenic uptake is 143 PgC over the same period.

### 5.3 Atmospheric $CO_2$

The land and ocean carbon fluxes have been put into two atmospheric tracers as described in Law et al. (2015, Sec. 2.4). These tracers have no impact on the model simulation but allow the atmospheric $CO_2$ distribution to be assessed. A reasonable simulation of known features of atmospheric $CO_2$ can increase our confidence in the simulated carbon fluxes. For example the seasonal cycle of atmospheric $CO_2$ is strongly driven by the seasonality in land carbon fluxes. Therefore, our simulated seasonality can be realistically compared to present day atmospheric $CO_2$ observations.

The seasonal cycle of atmospheric $CO_2$ is shown for four locations at different latitudes (Fig. 18, note the different vertical scale in the upper and lower panels). Seasonal cycles from the PresLAI and ProgLAI cases are calculated as the mean over the last 20 years of the historical period (1986-2005) with the annual mean removed from each year. The seasonality is plotted for the contribution from the land carbon fluxes only and for both the land and ocean carbon fluxes combined. The model output was taken from the nearest grid point to each location with the exception of Mace Head, where the model was sampled further west to better approximate the observations which are selected for clean-air (ocean) conditions.

As observed, the amplitude of the seasonal cycle decreases from north to south. At Alert ($82°$ N, Fig.18(a)) both model simulations overestimate the seasonal amplitude by up to 6 ppm with the growing season starting earlier than currently observed. The ocean carbon fluxes contribute little to seasonality at this latitude. At Mace Head ($53°$ N, Fig.18(b)) the simulated seasonal cycle is comparable to that observed with only a small difference in the seasonal amplitude (smaller than 2 ppm), while at Mauna Loa ($20°$ N,Fig.18(c)) the ProgLAI case better represents the observed seasonality than the PresLAI case.

Seasonal cycles in the southern hemisphere (e.g. South Pole) are more challenging to simulate correctly as they are made up of roughly equal contributions from local land fluxes, northern hemisphere land fluxes and ocean fluxes. Figure18(d) shows for the PresLAI case that the simulated seasonality from the land carbon fluxes is shifted in phase when the ocean carbon contribution is included but the phase shift is away from the observed seasonality. This phase shift is not apparent for the case with ProgLAI.

# 6  Conclusions

The evaluation of ACCESS-ESM1 over the historical period is an essential step before using the model to predict future uptake of carbon by land and oceans. Here, we performed two different scenarios for the evaluation of the land carbon cycle: running ACCESS-ESM1 with a prescribed LAI and a prognostic LAI. Running with a prognostic LAI is our preferred choice, since this includes the vegetation feedback through the coupling between LAI and the leaf carbon pool. However, results have shown that we overestimate the amplitude of the prognostic LAI annual cycle in the northern and southern hemisphere and underestimate it in the tropics. In future versions we need to improve the performance of the prognostic LAI, particularly for evergreen needle leaf and C4 grass.

ACCESS-ESM1 shows a strong cooling response to anthropogenic aerosols, which is offsetting the warming due to increases in greenhouse gases. The aerosol radiative forcing over the historical period is much stronger than the IPCC best estimate, but still within the uncertainty range. The impact of the cooling due to anthropogenic aerosols in ACCESS-ESM1 needs to be quantified in future work.

The land carbon uptake over the historical period is about $40\%$ larger for the run with prognostic LAI in comparison to the run with prescribed LAI. This is mainly due to the stronger response to volcanic eruptions which increases GPP in the tropics and reduces plant respiration globally, therefore increases NEE.

Globally integrated sea-air $CO_2$ fluxes are well captured and we reproduce very well the cumulative uptake estimate from the Global Carbon Project (Le Quéré et al., 2015) and our anthropogenic uptake agrees very well with observed GLODAP value of Sabine et al. (2004). The spatial distribution of sea-air $CO_2$ fluxes is also well reproduced by CMIP5 models and observations. At the same time global ocean NPP also shows good agreement with observations and lies well within the range of CMIP5 models. However, seasonal biases do exist in sea-air $CO_2$ fluxes and NPP, potentially related to biases in mixed layer depth and surface temperature that are present in ACCESS-ESM1; and will need to be addressed in later versions of ACCESS-ESM1.

Simulated carbon pool sizes are generally within the range of estimates provided in the literature. Simulated soil organic carbon has been compared against the Harmonized World Soil Database, finding very good agreement in the spatial distribution and the total size. Nitrogen and phosphorus limitation were active in our simulations and pool sizes seem reasonable if compared with other estimates. However, nitrogen and phosphorus cycles are poorly constrained and only a few global estimates exist with large uncertainties.

ACCESS-ESM1 has the capability of putting land and ocean carbon fluxes into tracers, which provides a way of assessing simulated atmospheric $CO_2$ concentrations. The simulated seasonal cycle is close to the observed, but we overestimate the amplitude in the high northern latitude by up to $6\,\mathrm{ppm}$ and we also notice small phase shifts.

Overall, land and ocean carbon modules provide realistic simulations of land and ocean carbon exchange, suggesting that ACCESS-ESM1 is a valuable tool to explore the change in land and oceanic uptake in the future.

**Code availability**

Code availability varies for different components of ACCESS-ESM1. The UM is licensed by the UK Met Office and is not freely available. CABLE2 is available from https://trac.nci.org.au/svn/cable/. See https://trac.nci.org.au/trac/cable/wiki/ CableRegistration for information on registering to use the CABLE repository. MOM4p1 and CICE are freely available under applicable registration or copyright conditions. For MOM4p1 see http://data1.gfdl.noaa.gov/~arl/pubrel/r/mom4p1/src/ mom4p1/doc/mom4_manual.html. For CICE see http://oceans11.lanl.gov/trac/CICE. For access to the MOM4p1 code with WOMBAT as used for ACCESS-ESM1, please contact Hailin Yan (Hailin.Yan@csiro.au). The OASIS3-MCT 2.0 coupler code is available from http://oasis.enes.org.

*Acknowledgements.* This research is supported by the Australian Government Department of the Environment, the Bureau of Meteorology and CSIRO through the Australian Climate Change Science Programme. The research was undertaken on the NCI National Facility in Canberra, Australia, which is supported by the Australian Commonwealth Government. The authors wish to acknowledge use of the Ferret program for some of the analysis and graphics in this paper. Ferret is a product of NOAA's Pacific Marine Environmental Laboratory. (Information is available at http://ferret.pmel.noaa.gov/Ferret/). We acknowledge the World Climate Research Programme's Working Group on Coupled Modelling, which is responsible for CMIP, and we thank the climate modelling groups (listed in Table 1 of this paper) for producing and making available their model output. For CMIP the U.S. Department of Energy's Program for Climate Model Diagnosis and Intercomparison provides coordinating support and led development of software infrastructure in partnership with the Global Organization for Earth System Science Portals.

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

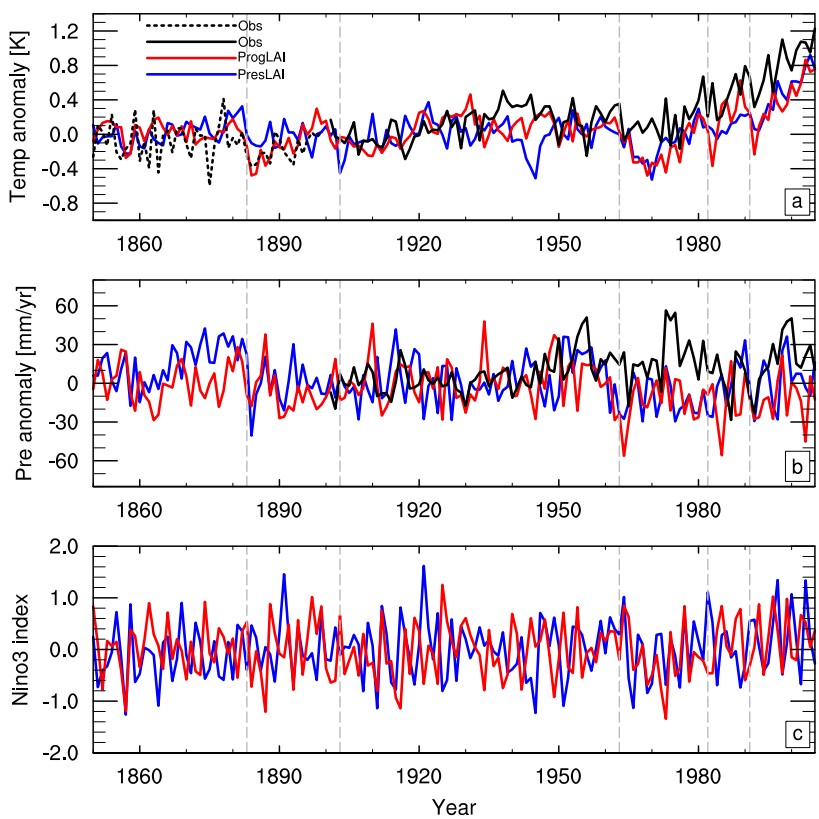

**Figure 1.** Anomalies (reference period: 1901-1930) for **(a)** globally averaged surface air temperature and **(b)** globally averaged precipitation for land points only for ACCESS-ESM1 (PresLAI, blue; ProgLAI, red) and observed CRU (black, dashed before 1901). Major volcanic eruptions are marked with dashed lines: Krakatoa (1983), Santa Maria (1903), Mt. Agung (1963), El Chichón (1982) and Mt. Pinatubo (1991).

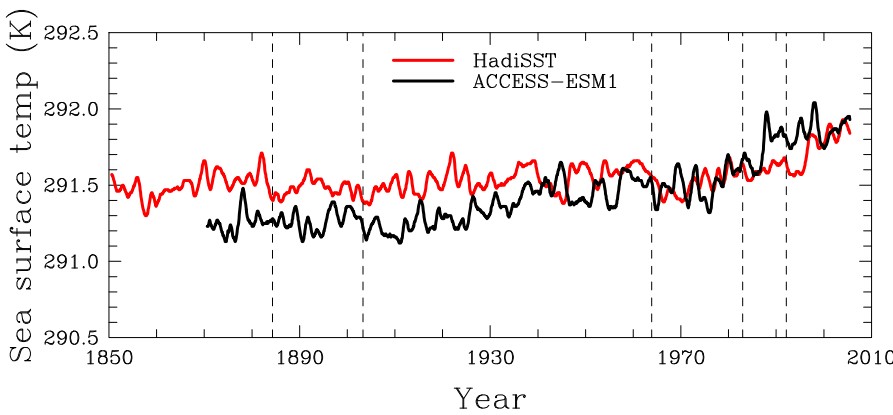

**Figure 2.** Globally averaged sea surface temperature (K) between 1850- 2005, red is ACCESS-ESM1 and black is HadiSST (Rayner et al., 2003). Major volcanic eruptions are marked with dashed lines: Krakatoa (1983), Santa Maria (1903), Mt. Agung (1963), El Chichón (1982) and Mt. Pinatubo (1991).

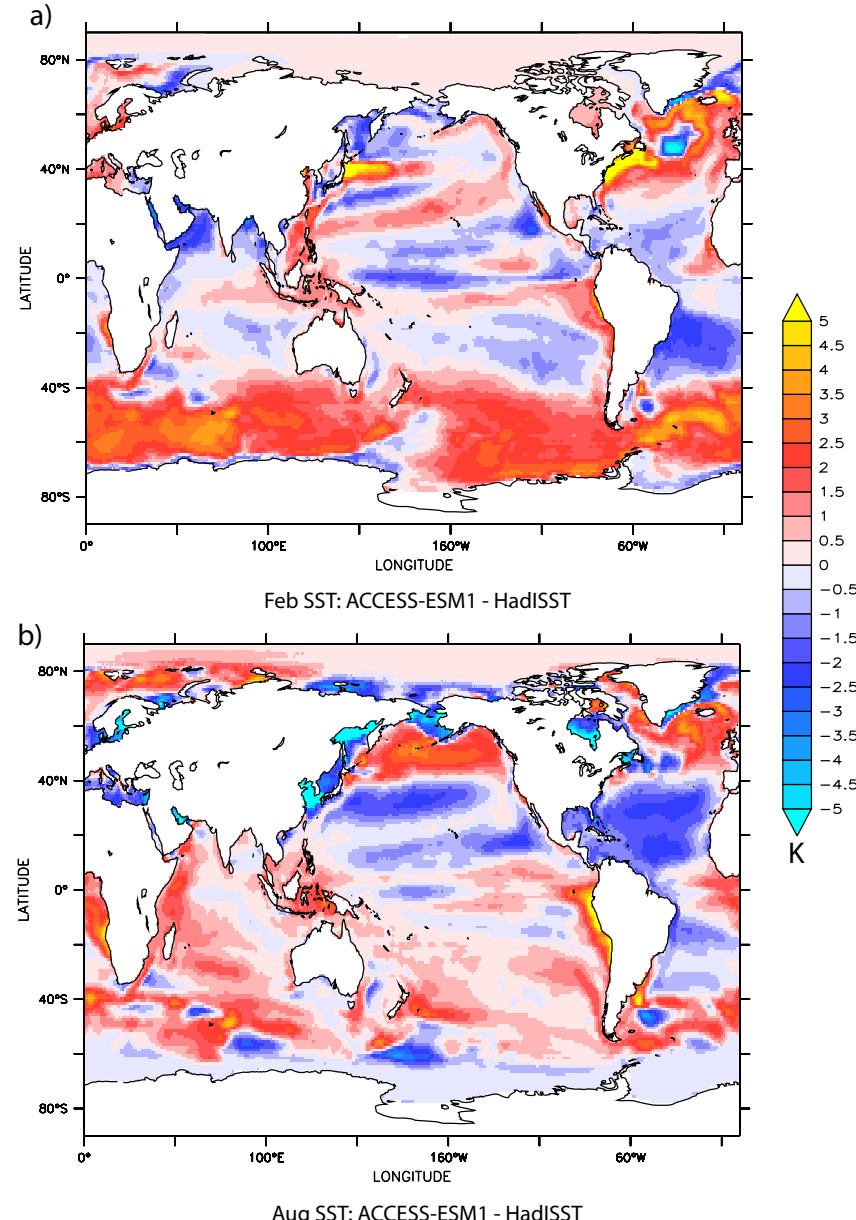

**Figure 3.** Differences in sea surface temperature (K) between ACCESS-ESM1 and HadiSST for **(a)** February and **(b)** August.

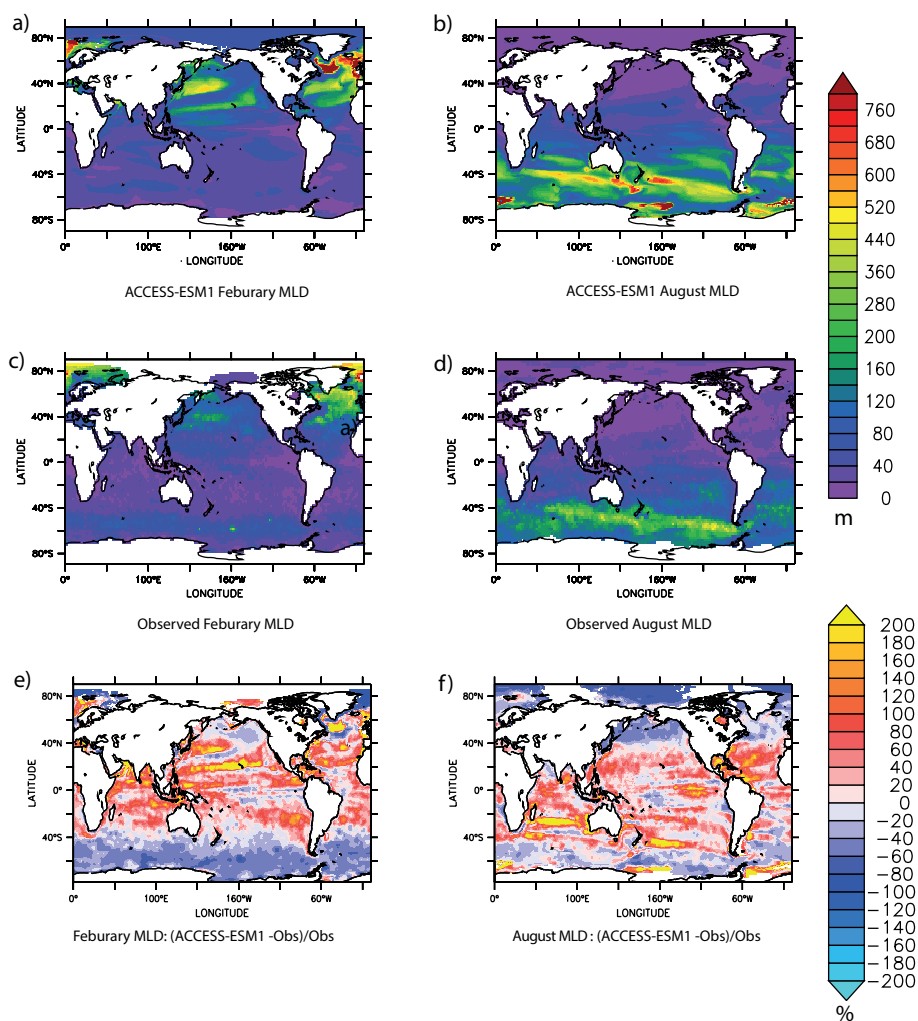

**Figure 4.** Differences in mixed layer depth between ACCESS-ESM1 and observations de Boyer Montégut et al. (2004) for **(a,c)** February and for **(b,d)** August. Panels **(e,f)** show the percentage difference between de Boyer Montégut et al. (2004) and ACCESS-ESM1 calculated as ((OBS–ACCESS-ESM1)/OBS)*100. The mixed layer is calculated based on a $0.03 \, \mathrm{kg \, m^{-3}}$ density change from the surface ocean.

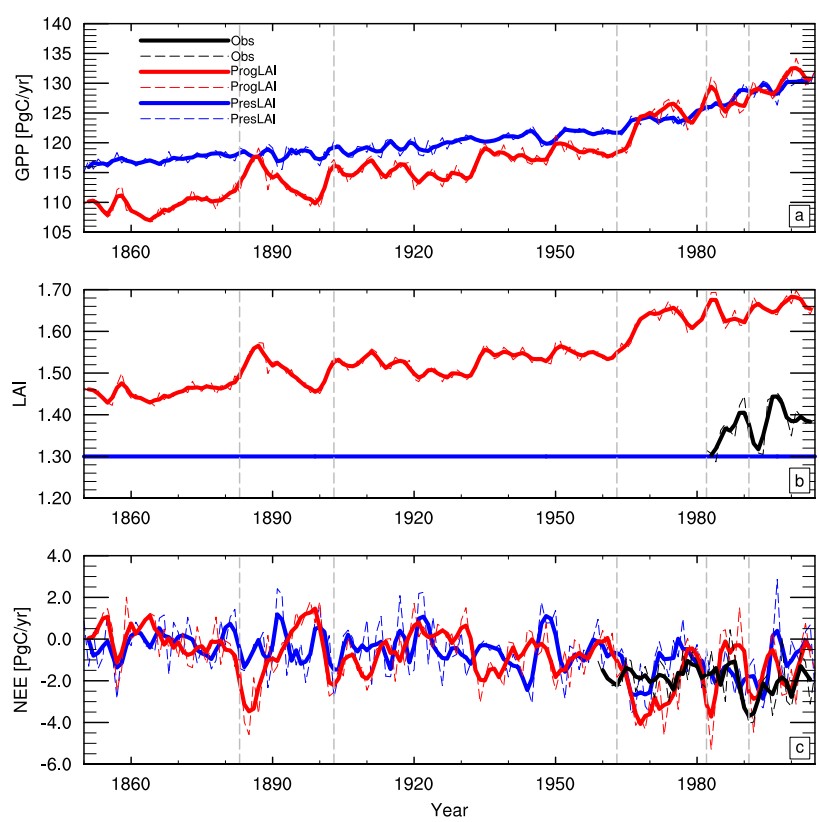

**Figure 5.** Temporal evolution of **(a)** GPP ($PgC\,yr^{-1}$), **(b)** LAI and **(c)** NEE ($PgC\,yr^{-1}$). GCP estimates for NEE are shown for comparison in black for the years 1959-2005. ACCESS-ESM1 results are shown for PresLAI (blue line) and ProgLAI (red line) with annual values marked in thin dashed lines and a 5 yr running mean in heavy solid lines. Major volcanic eruptions are marked with dashed lines: Krakatoa (1983), Santa Maria (1903), Mt. Agung (1963), El Chichón (1982) and Mt. Pinatubo (1991).

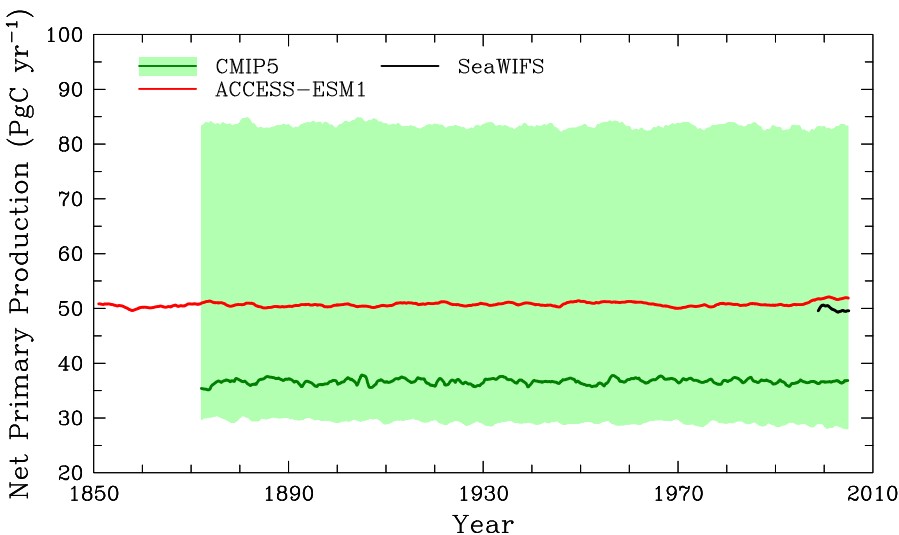

**Figure 6.** Comparison of Integrated Net Primary Production ($PgC\,yr^{-1}$) in the period 1850-2005 between CMIP5 and ACCESS-ESM1. The solid red line represents the integrated carbon uptake in $PgC\,yr^{-1}$ from ACCESS-ESM1, while the green line represents the median of the CMIP5, model with the range overlain (as shaded area) as the 10th and 90th percentiles. Overlain on this plot are the observed values from SeaWIFS over the period 1998-2005 in black.

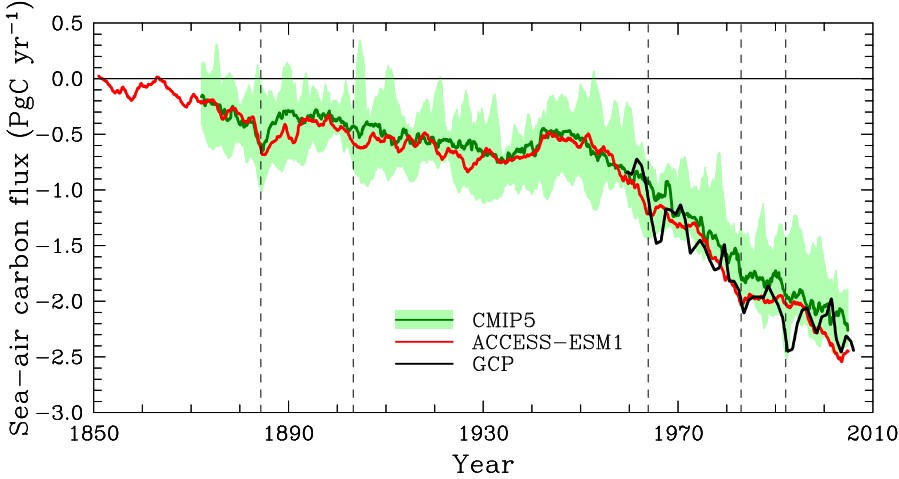

**Figure 7.** Comparison of sea-air $CO_2$ fluxes ($PgC\,yr^{-1}$) in the period 1850-2005 carbon uptake from ACCESS-ESM1. The solid green line represents the median of the CMIP5, while the shaded are represents the 10th and 90th percentiles of the CMIP5 model. Overlain on this is the estimated sea-air fluxes from the Global Carbon Project (Le Quéré et al., 2015) in black; and the timing of major volcano eruptions over the historical period.

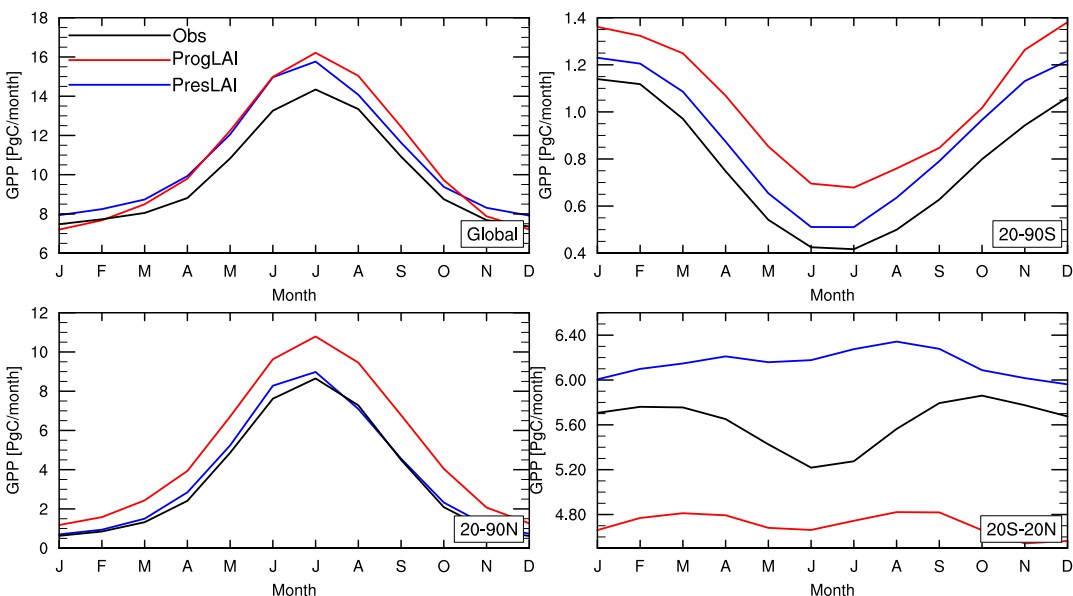

**Figure 8.** Mean annual cycle of GPP ($\mathrm{PgC\,month^{-1}}$) for the period 1986-2005. ACCESS-ESM1 results are shown in blue (PresLAI) and red (ProgLAI). Observation based estimates are shown in black.

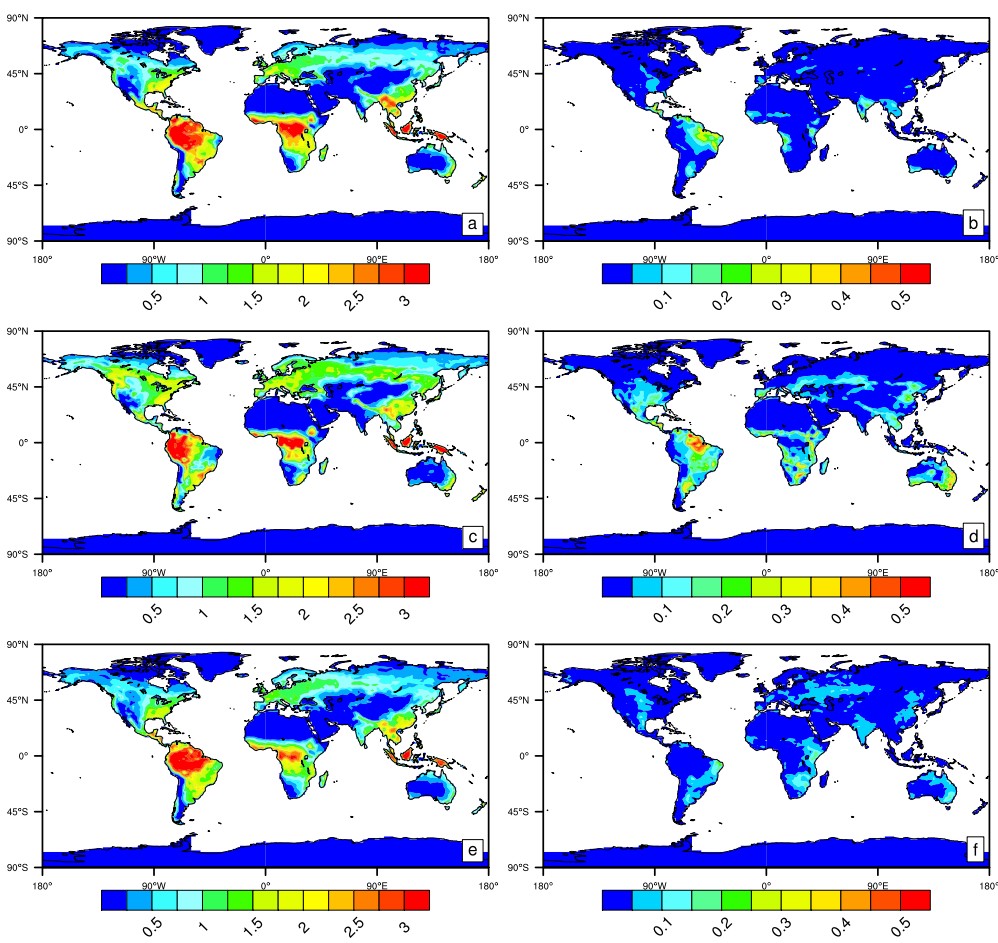

**Figure 9.** Spatial distribution of **(a,c,e)** GPP and **(b,d,f)** GPP IAV ($\mathrm{kgC\,m^{-2}\,yr^{-1}}$) for **(a,b)** PresLAI, **(c,d)** ProgLAI and **(e,f)** observation based estimates.

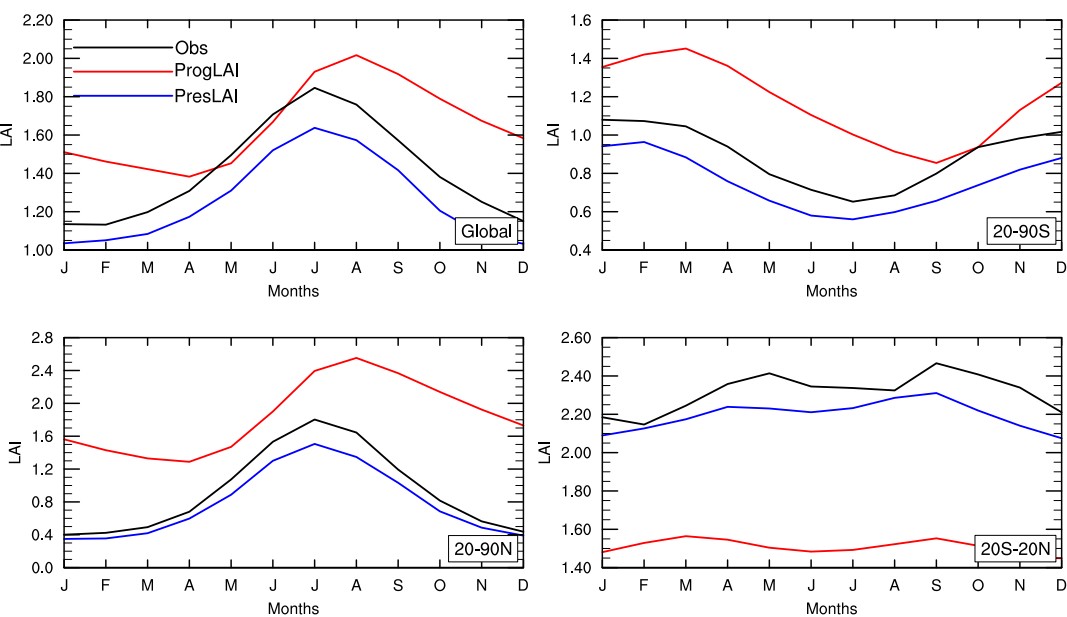

**Figure 10.** Mean annual cycle of LAI for the period 1986-2005. ACCESS-ESM1 results are shown in blue (scenario with prescribed LAI) and red (scenario with prognostic LAI). Observation based estimates are shown in black.

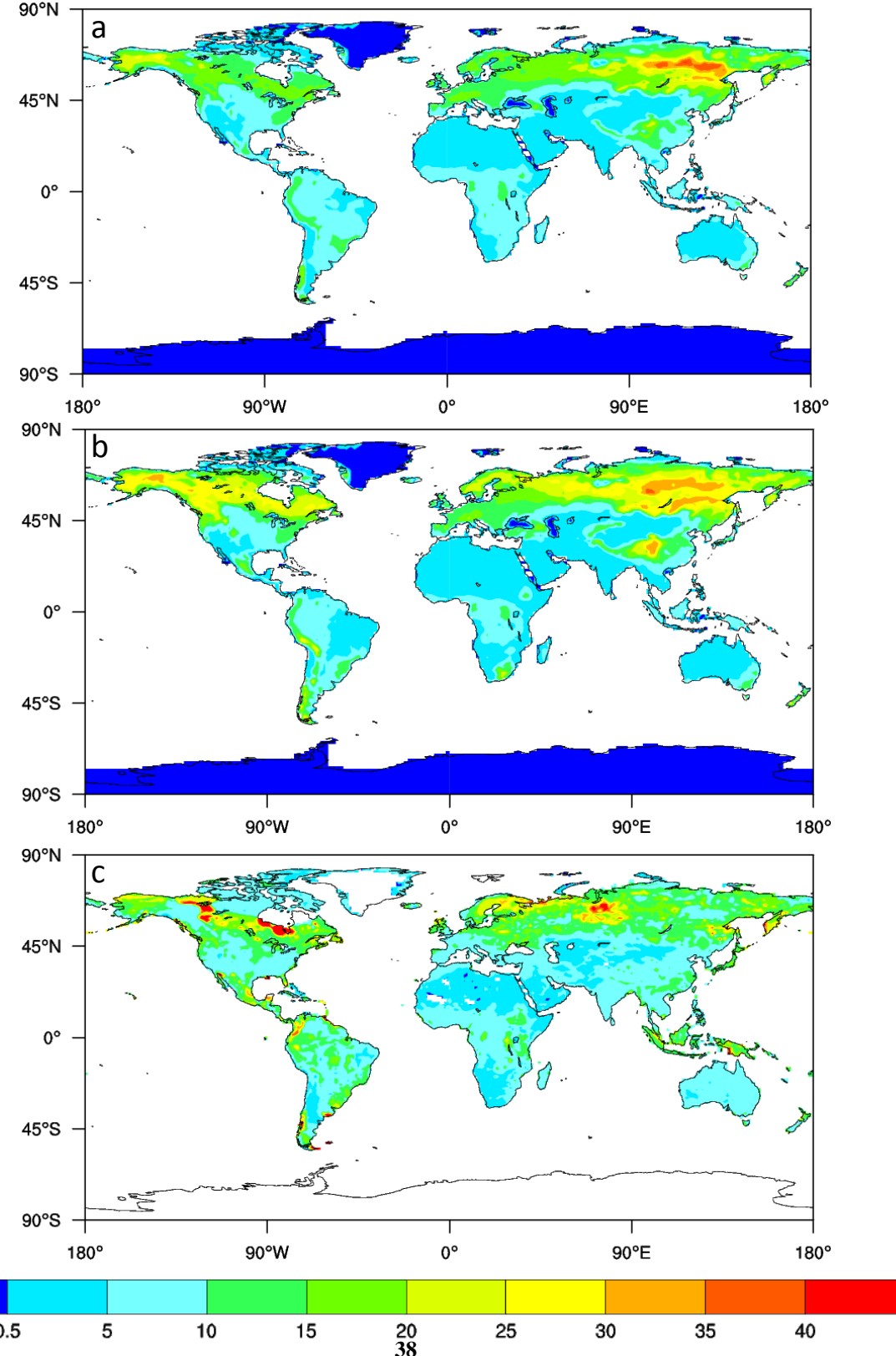

**Figure 11.** Spatial distribution of organic soil carbon ($kgC\,m^{-2}$) (a) using prescribed LAI, (b) using prognostic LAI and (c) observation based estimated from HWSD.

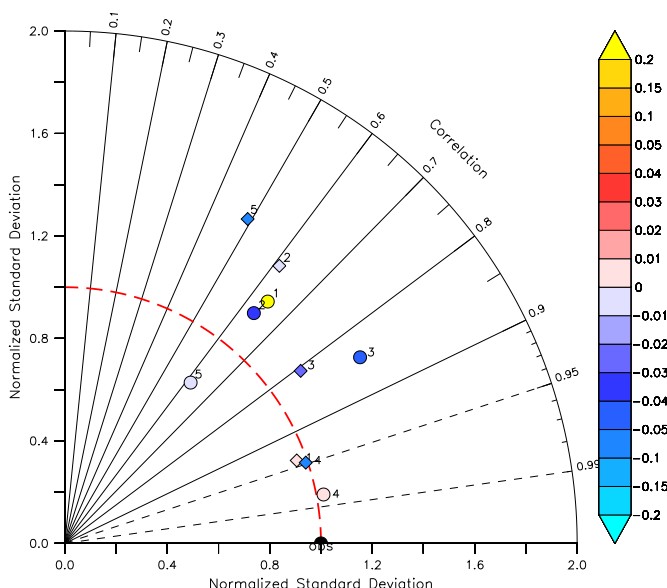

**Figure 12.** Taylor diagram assessing the response of the ACCESS-ESM1 simulations (circles), and the median of CMIP5 models (diamonds) with observations. The numbers correspond to: (1) Phosphate, (2) Alkalinity, (3) DIC, (4) SST, and (5) (sea surface) Salinity. For explanation of how to intepret the diagram please see the text.

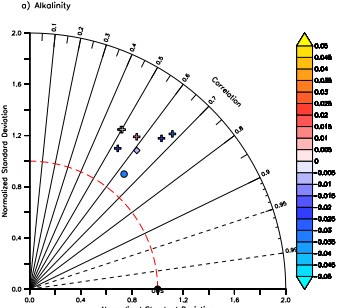 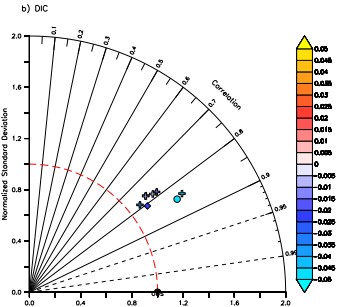

**Figure 13.** Taylor diagram assessing the alkalinity **(a)** and DIC **(b)** of the ACCESS-ESM1 simulation (circle), the median of CMIP5 models (diamond), and the individual members of the CMIP5 ensemble (crosses) with observations.

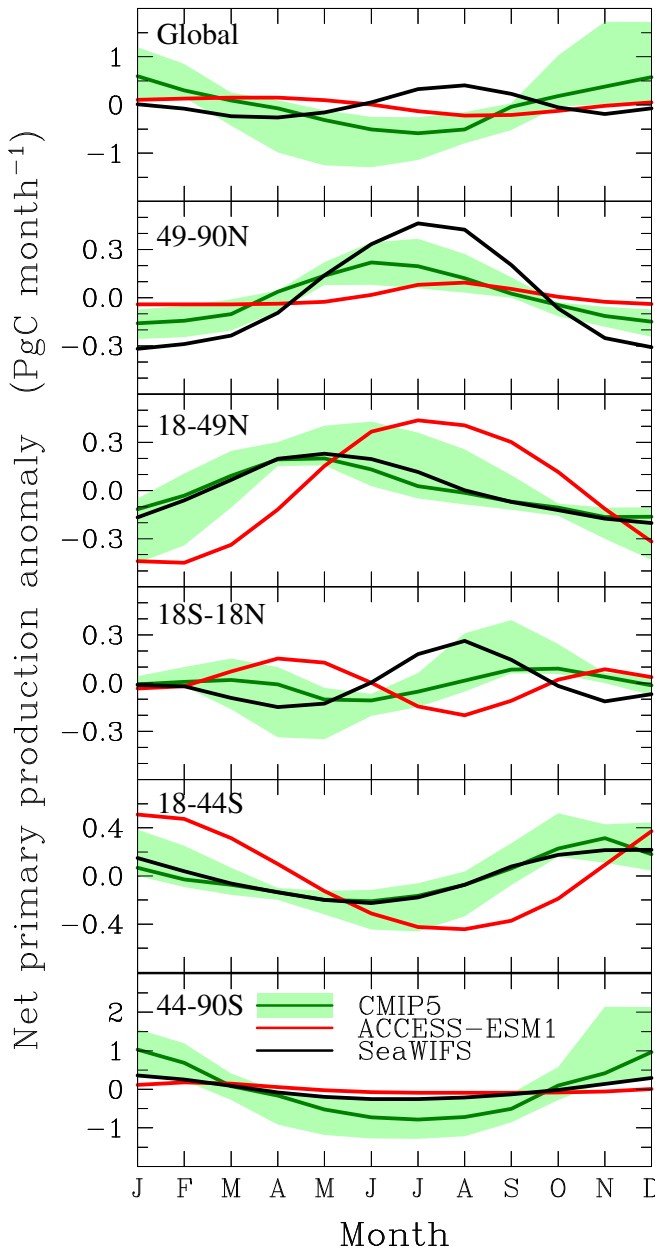

**Figure 14.** The seasonal cycle of NPP anomalies $(\mathrm{PgC\,month^{-1}})$ from ACCESS-ESM1 in red and SeaWIFS (Behrenfeld and Falkowski, 1997) in black calculated over the period 1998-2005. Overlain on this plot is the CMIP5 the median (solid green line) and the range 10th and 90th percentiles (shaded).

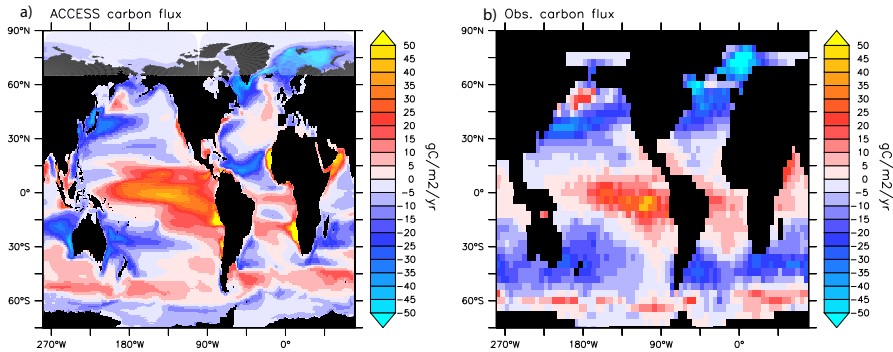

**Figure 15.** The integrated sea-air $CO_2$ fluxes over the period 1986-2005 from **(a)** ACCESS-ESM1 and **(b)** Wanninkhof et al. (2013).

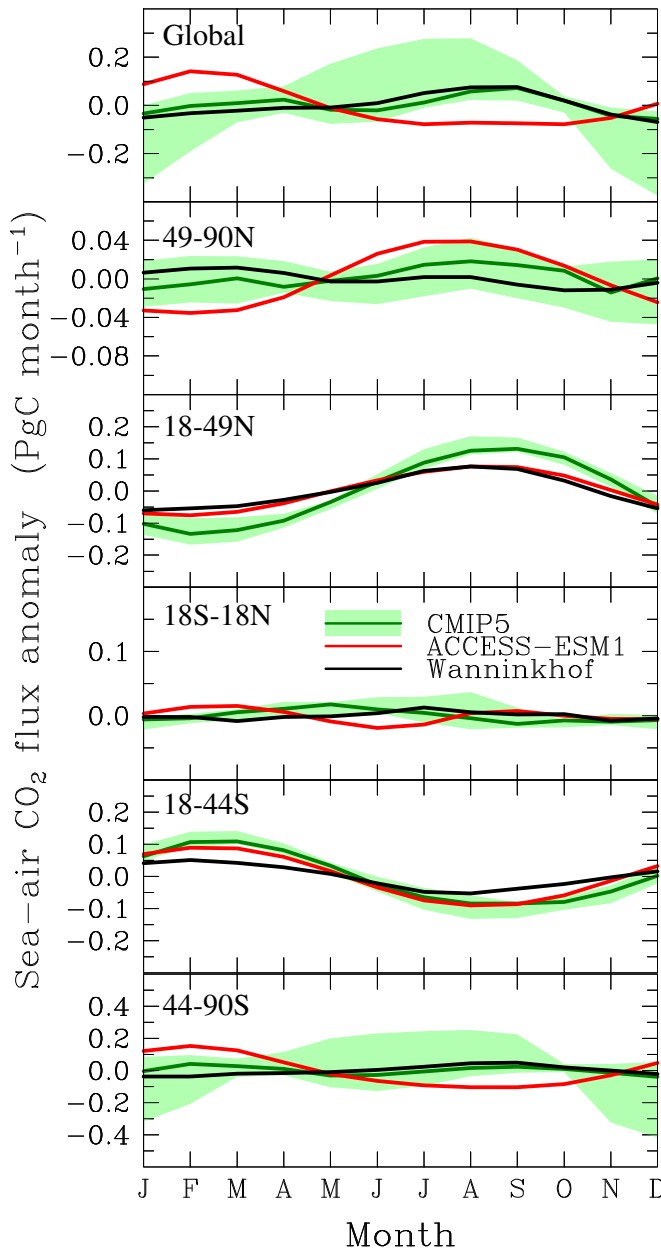

**Figure 16.** The seasonal cycle (1986-2005) of sea-air $CO_2$ flux anomalies ($\mathrm{PgC\,month^{-1}}$) from ACCESS-ESM1 (red line) and observations ((Wanninkhof et al., 2013); black line). Overlain is the CMIP5 median (solid green line) and the range as the 10th and 90th percentiles (shaded).

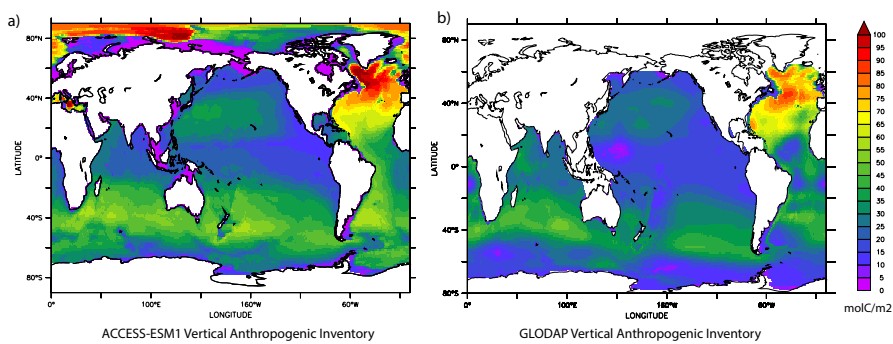

**Figure 17.** Column inventory of Anthropogenic Carbon in the ocean $(\mathrm{molC\,m^{-2}})$ from **(a)** ACCESS-ESM1 and from **(b)** GLODAP (Key et al. (2004) for 1994.

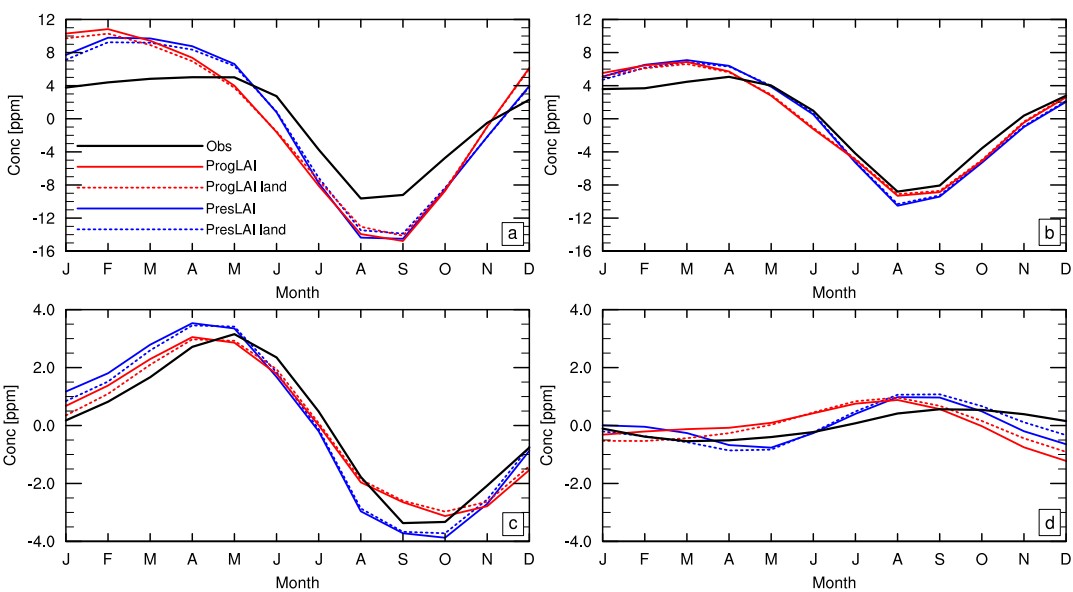

**Figure 18.** Mean seasonal cycle of atmospheric $CO_2$ for the period 1986-2005 from land carbon fluxes (dashed lines) and both land and ocean carbon fluxes (solid line). The prescribed LAI case is shown in blue, the prognostic LAI case in red and observations based on flask data from GLOBALVIEW in black for **(a)** Alert (82.45° N, 62.52° W), **(b)** Mace Head (53.33° N, 9.90° W), **(c)** Mauna Loa (19.53° N, 155.58° W) and **(d)** South Pole (89.98° S, 24.80° W).

**Table 1.** The CMIP5 models used to assess the ocean response of ACCESS-ESM1 over the historical period in the study. Reference for all models are provided in Lenton et al. (2015).

| Model Name | Institute ID | Modelling Group |
|---|---|---|
| CanESM2 | CCCMA | Canadian Centre for Climate Modelling and Analysis |
| HadGEM-ES | MOHC | Met Office Hadley Centre (additional HadGEM2-ES |
|  | (additional realizations by INPE) | realizations contributed by Instituto Nacional de Pesquisas Espaciais) |
| GFDL-ESM2M | NOAA GFDL | NOAA Geophysical Fluid Dynamics Laboratory |
| ISPL-CM5A-LR | IPSL | Institut Pierre-Simon Laplace |
| IPSL-CM5A-MR | IPSL | Institut Pierre-Simon Laplace |
| MPI-ESM-MR | MPI-M | Max-Planck-Institut für Meteorologie |
|  |  | (Max Planck Institute for Meteorology) |

**Table 2.** Mean carbon (C), Nitrogen (N) and phosphorus (P) pools sizes in Pg for pre-industrial (780-799) and present day (1986-2005). Historical changes (1850-2005) for C are also shown. Biomass comprises leaf, wood and root pool.

| | Pre-industrial | | | | | | Present day | | | | | | Historical change C | |
|---|---|---|---|---|---|---|---|---|---|---|---|---|---|---|
| | PresLAI | | | ProgLAI | | | PresLAI | | | ProgLAI | | | PresLAI | ProgLAI |
| Pool | C | N | P | C | N | P | C | N | P | C | N | P | ΔC | ΔC |
| Biomass | 611 | 5.7 | 0.31 | 731 | 6.15 | 0.33 | 670 | 6.2 | 0.34 | 807 | 6.8 | 0.37 | 69.5 | 87.2 |
| Litter | 117 | 0.85 | 0.04 | 149 | 1.02 | 0.05 | 126 | 0.9 | 0.05 | 163 | 1.1 | 0.06 | 7.6 | 12.3 |
| SOC | 1034 | 82 | 9.6 | 1187 | 86.1 | 11.9 | 1050 | 83.4 | 10.1 | 1217 | 88.5 | 12.6 | 20.5 | 37 |
| ∑ | 1762 | 88.6 | 10.0 | 2067 | 93.3 | 12.3 | 1846 | 90.5 | 10.5 | 2187 | 96.4 | 13.0 | 97.6 | 136.5 |