# Peer review of "The carbon cycle in the Australian Community Climate and Earth System Simulator (ACCESS-ESM1). 2. Historical simulations"

_Geoscientific Model Development, 2016_

## Referee Comment (RC1) · C.D. Jones (Referee) · 7 Jun 2016

Review of "The carbon cycle in the Australian Community Climate and Earth System Simulator (ACCESS-ESM1). 2. Historical simulations", by T Ziehn et al.

This is a partner manuscript (part 2) which together with Law et al Part-1 document and describe the configuration and performance of the ACCESS Earth System model. My remit clearly is to review this paper, but I make a couple of comments which probably span both papers. This is exactly (in my view) the ideal paper for GMD and the authors deserve credit for a thorough job.

Overall I find them to be a comprehensive and well written pair of papers and that the

model is clearly well formulated and calibrated and performs well in both pre-industrial and historical simulations. There would be no problem at all in including this model in multi-model comparisons with other CMIP5 models. The two main issues I would encourage the authors to address in the run-up to CMIP6 are to do with spin-up and land-use - and I know both of these are in progress. Neither require major changes to the paper but could be mentioned (the latter is covered well actually but the former needs more discussion I think - see detailed comments below).

Otherwise I have only very minor comments that may improve clarity and also a few suggestions for other evaluation activities the authors may consider, but are not required for publication of this paper.

Chris Jones.

—

Major comments

1. My main concern with this model is the length of time taken to achieve a spun-up state. Law et al document this nicely, but I think it requires more discussion in this paper too what the implications are. The drifts in carbon stores are still non-negligible even after 800 years of spin-up (your start point here). I think this should be laid out explicitly before the analysis starts. You do, in the land carbon section, acknowledge this and subtract the control run drift. But unless a reader has been through Law et al they would not know how big this drift is. For the ocean it is more important still, and the ocean section does not mention this at all. The drift of circa 0.7 GtC/yr (figure 11a in Law et al) is of similar magnitude to your historical fluxes (I assume these are corrected for the drift). If a reader hadn't seen that figure then they would not realise from this paper the size of drift being subtracted.

In CONCENTRATION-driven runs like this, you can of course force the correct CO2 and correct for the drift after the simulation. But that is not possible in an EMISSION-

driven run, and such a drift would cause a massive drift in atmospheric CO2 rendering an emissions-driven historical run meaningless. This would, at present preclude use of this model in C4MIP for example which would be a great shame. The latest C4MIP protocol (Jones et al 2016, GMDD - CMIP special issue) recommends a maximum acceptable drift of 10 GtC per century. I therefore throughly recommend that ACCESS modellers attempt to find accelerated means to derive a spun-up state in time for CMIP6. There are numerous options, such as running offline (for either land or ocean), or using reduced turn-over time techniques as per Koven et al for CLM (http://www.biogeosciences.net/10/7109/2013/bg-10-7109-2013.pdf).

as a final word on this, lack of carbon conservation would also be more of an issue for E-driven runs.

2. My second concern is the lack of land-use change as a forcing. You already know this, and acknowledge it in the paper, so no revisions to the manuscript are required, but I just take this opportunity to stress that simulations of contemporary and future climate/carbon cycle are very much reduced in usefulness if they lack the very large land-use forcing of the land carbon cycle. Implementing this for CMIP6 is also therefore a priority I would say.

Minor comments

1. Having quickly read Law et al before I reviewed this one I was struck that there was not an evaluation there of (land) carbon stocks. I do feel that the land carbon modelling community have become fixated on evaluation against fluxes to the detriment of stocks and residence times. This is beginning to change and I was pleased to see some discussion of carbon stocks in this paper. It would be nice to see the time changes in these though as well - could table 2 be extended to show pre-industrial and present day stocks? In the discussion on biomass you mention that your results are higher than observational based estimates - but of course you lack land-use change as a driver. So it could easily be expected that your biomass is of the order 100-150 GtC too high

due to this. If you masked for present day agricultural regions you would probably get a much closer fit to expected global totals. So your simulation is actually not bad.

2. Abstract - you can say that aerosol forcing is large or larger than other models. But don't say "over-sensitive" as we simply don't know. Maybe this is correct. . .

3. section 2.3 - I like the comparison vs CMIP5 models. This is a nice way to put model-data discrepancies in context. But why do you only include 5 CMIP5 models (counting the 2 IPSLs as one model). Anav et al used more than twice that - any reason not to use the full set?

4. in a couple of places (e.g. start sec 3.1) you mention the variability of the land sink and/or the atmospheric CO2. You could go further and use this as an evaluation metric. Both Cadule et al (2010, GBC) and Cox et al (2013, Nature) show the power of the C-cycle sensitivity to ENSO on inter-annual timescales as a really strong evaluation metric and constraint on the sensitivity of ESMs.

5. p.7, line 21. I don't understand why you attribute your slower warming to the initial warm bias - how do you know the warm bias causes this? Could just be under-sensitive SSTs not related to a bias.

6. p.8 line 3. I was amused to see that having an error different from other models was "encouraging"! not sure why! Can you say why it is better to have the opposite MLD bias from other models?

7. in general I thought this MLD section (3.2) was a bit superficial - can you be a bit more quantitative in your comparison and description? the figure shows the data but it can be hard to tell from there if the differences you describe are of the order of a few % or 10s of % or factors of 2 or more or what? And can you mention your confidence in the ons? presumably global maps of MLD are not directly observed but must have certain extrapolation uncertainties and so on. Are some areas/seasons of the oceans better sampled than others etc. . .

8. p. 8 line 21. Can you define what you mean by IAV. Interannual Variability I know, but how do you turn this into a number? is it the standard deviation of a time series of annual means? in which case is the time series de-trended first? etc

9. p.9 When discussing historical changes in land carbon can you split into veg and soil changes (e.g. put in table 2). You could compare directly with the 2 models in Jones et al (2013) which also don't have land-use forcing (dashed lines in figure 2). You could probably also compare with model results from detection-and-attribution studies which ran with/without certain forcings. The are probably various no-land-use runs to look at.

10. Sec 4.2. Despite the large drift your historical ocean sink does look a very close match to the obs. Can you also quote a cumulative uptake here?

11. sec 5.1.2. I was curious to see that your prescribed LAI didn't match your observations. Can you explain why not? You illy that this is because there are differences between observed datasets - which is of course true. But then why do you choose one dataset to prescribe LAI to the model, and then a different one to evaluate against? If one is better than the other can you use it for both?

12. p.13 discussion of different carbon stocks for the two configurations. Given both have very similar GPP, the large difference in biomass is presumably due to residence times? Having GPP further north in compared to the tropics means that for the same global GPP you have a higher biomass? You might consider next time an evaluation of turnover times - e.g. as per Carvalhais et al (2014, Nature).

13. section 5.2.1 / 5.2.2. Can you swap the order of figures 12 and 13? you describe 13 first.

14. sec 5.2.3. You compare global totals between model and GLODAP, but the map shows missing areas in the ons dataset. So should you first mask out the model to match the same area before you compare totals? (or quote a full AND a masked number for the model). The values in the Arctic for example are very high but missing

in the obs (although of course the area is smaller than this projection makes it look).

15. sec 5.3. It seems reasonable that the land drives most of the seasonal cycle at your ons stations except the south pole. I like the way you have labelled the land and ocean CO2 separately to be able to diagnose this. But I couldn't work out why the contribution of the ocean at the south pole looks different for your two LAI configurations. If I read the figure correctly then the role of the ocean is given by the blue solid minus the blue dashed line and the red solid minus the red dashed line. These are quite different by eye - e.g. the blue lines are quite far apart in December and the red lines quite close. So why does your LAI treatment affect your December ocean fluxes so much?

16. Table 2. Can you add a final row at the bottom of "total"

17. general comment on figures - maybe a personal preference but please can you add a legend so that I can easily read which line is which. The text in the captions is very good - but I have to read a whole paragraph to spot what the red/blue lines are. Would be great to simply see a legend with this in as well as the detail in the caption text.

---

## Referee Comment (RC2) · Anonymous Referee #2 · 13 Jun 2016

**Review:** The carbon cycle in the Australian Community Climate and Earth System Simulator (ACCESS-ESM1). 2. Historical simulations.

**Authors:** T. Ziehn, A. Lenton, R.M. Law, R.J. Matear, and M.A. Chamberlain

**Date:** 12 June, 2016

**General Remarks:**
This paper evaluates the carbon cycle in ACCESS-ESM1, comparing historical simulations against both CMIP-5 model results and observations. The evaluations presented in this paper highlight both strengths and weaknesses in the ACCESS-ESM1 carbon cycle. The comparisons displayed and discussed document the baseline behavior of ACCESS-ESM1, which is essential for model justification needed by future studies utilizing this tool. Overall, this paper accomplishes its goal of showing that the ACCESS-ESM1 is a useful tool in exploring the carbon cycle, including both land and oceanic fluxes and carbon storage.

This paper contributes to modeling science by evaluating and documenting the carbon cycle modeling capabilities in ACCESS-ESM1. In general, this study is structured well and presented in a clear fashion, and I found the description to be sufficient to satisfy the reproducibility requirements. While the scientific approach and applied methods are valid, I found the text to be too subjective rather than quantitative. Instead of using generalized, generic comments on the model performance, I feel that this paper requires more scientific and statistical-based evaluations in many of the presented comparisons. Modifications to include more direct, quantitative and supportable statements would not only improve the scientific quality of the text, but would also strengthen the validity of the model evaluation. Despite this criticism, I feel this study is suitable for publication in Geoscientific Model Development with revisions to address this concern.

**Specific Comments:**
The main focus of these specific comments is to highlight text that could benefit from being more quantitative rather than subjective. In addition, there are some specific suggestions for clarifications aimed at improving the overall flow and clarity of the manuscript. Before starting the specific suggestions, one other general remark I have is that the figure axes and color bar labels are very small and hard to read.

**Abstract**
The abstract provides an overall view of how well ACCESS-ESM1 performs; however, they provide this information using terms such as "good" and "performs well". How well? Good according to whom? Is your good the same as my good? First off, the authors state ACCESS-ESM1 overestimates the seasonal amplitude of LAI, but do not attach any quantities to this statement. Rather than providing a quantitative assessment of this statement, the reader is left to wonder if this is a substantial bias, perhaps even being prohibitive of using this model in the future, or merely a relatively minor difference that is offset by other positive features. The paper then continues with the statement that the oceanic and land fluxes "show good agreement with the observations", but again, no metric is used. How big are the differences, what is the error, or how closely correlated, both spatially and temporally, are they to observations and/or other CMIP-5 models? In the last overview comparison, the authors state that the seasonal cycle is "close to the observed seasonal cycle", but as the reader I have to wonder how close is close? I believe that putting quantifiable

metrics on at least some of these statements will strengthen the concluding remark that ACCESS-ESM1 is indeed a useful tool.

**Observations**
The model evaluation (later in the text) proceeds through a straightforward succession of comparisons; however, the order of these comparisons is different than the order the data is listed in.  For clarity and consistency, it would be an easy fix to reorganize this section to present the data in the order that they are used.

**Land Temperature and Precipitation**
This section presents a time-series comparison of temperature and precipitation, but contains very little quantitative statements.  For example, the temperature anomalies are "close to the observed anomalies through most of the period".  Again I have to wonder, how close is close?  Perhaps a difference plot showing the errors would be useful?  Or perhaps a correlation coefficient that may or may not be significant?  Some sort of metric on this statement would be much more enlightening to the reader.   For example, as I look at Fig. 1 myself, I see the author's point that ACCESS-ESM1 is lower than the observations 1965-2005; however, to my eye I also see the model looks quite a bit lower in the 1940s.  I think that a difference plot would help identify these areas, rather then relying on the reader to have to assess the differences between the models and the observations on eyesight alone.  Another example later in this section occurs in the precipitation anomaly discussion: the authors state the differences are "generally small," but provide no values to suggest what is meant by small.  This is followed by the statement that "the simulations compare well with observed rainfall anomalies until about 1950," with no supporting metric such as minimal differences or significant correlation coefficients to back up this statement.  Overall, I found that this entire section had very few quantitative comparisons, instead relying heavily on subjective terminology, making me believe that at least some measurable metrics would improve the paper.

The last paragraph in this section discusses the timing of precipitation anomalies versus volcanic eruptions.  The authors point out a reduction in precipitation following eruptions, with the one exception of El Chichon; however, when I looked at the time-series, I did not see a decrease after the Santa Maria eruption in addition to El Chichon.  In fact, looking at the time-series, the decrease after volcanic eruptions did not stand out to me, especially when the authors note the reduction following the Mt. Pinatubo eruption, which does stand out, is too far away from the eruption date to be related.  I think a more quantifiable analysis of the magnitude and the timing of this decrease, in days or weeks or some stated time-scale, would be helpful.

**Sea Surface Temperature and Mixed Layer Depth**
The second full paragraph of this section discusses spatial patterns of sea surface temperatures, except no time period for the analysis is provided, not even in the caption for Figure 3.  Are these differences over the entire simulation or a selected time period?  Shifting to the text, the authors state that ACCESS-ESM1 "produces very heterogeneous differences from observations."  Reading this, it was unclear to me what was meant.  When I turned to look at the figure I expected to see random errors; however, in my opinion the differences are spatially coherent in latitude bands.  Then in this same discussion they state "there do not appear to be strong seasonal biases," but with this terminology I have to wonder are there or aren't there?  Then they state the exception of the North Atlantic, which has a coherent bias towards cooler temperatures, but to me it looks like this is a year-round bias more than a seasonal bias.  It perhaps does vary in magnitude with season, being a larger bias in August, but it looks like the sea surface temperature here is underestimated year round.  Further looking at Figure 3, I also see a flip-flop in errors in the southeastern Pacific Ocean,

with positive differences in August and negative differences in February. To me, having differences that vary with time of year makes it seasonal, but this region is not mentioned. I think this section could benefit from more careful wording and analysis.

The ocean mixed layer depth discussion could also benefit from a more quantitative analysis, rather than using statements like "appears to slightly overestimate the depth in winter" and "appears to underestimate the depths in summer," with no statistical support for these subjective comments. Also, in Figure 4 in this section, the caption states that differences are shown, suggesting a difference plot; however, the figures just show the results from both the model and the observations. I would suggest rewording the caption to avoid confusion.

**Land Carbon Response**

I think it would be helpful to include the MODIS/AVHRR LAI data in Figure 5b. I realize this would only be for the last few years of the simulation, but the benefit is that it would provide a reference to the simulated LAI values. After reading it through, I discovered there is a section on LAI where it is discussed in more detail, but this figure comes first, and when I read it I was wondering how it compared.

I found the discussion in the last two paragraphs of this section confusing. I loved to see values and uncertainties; however, it was unclear to me what values were comparable. The section starts with a discussion on carbon uptake, which at first I assumed represents total uptake, or GPP. But from what I could tell, the same values provided for ACCESS-ESM1 were then used in the following paragraph, which talks about NEP. I read the section several times, but only found the one value of 154 Pg C for prognostic ACCESS-ESM1. I was then confused when it was stated that this value is "at the low end of the CMIP5 range," when that range is estimated to be from -59 to 18 PgC according to Shao et al. (when outliers are not included) or from -124 to 50 PgC from Jones et al. Based on these numbers, doesn't ACCESS-ESM1 take up and store much more carbon (+154 Pg C) than the CMIP5 models? It doesn't help that the signs make this analysis even more confusing. Since NEP and uptake were discussed, I assumed a negative value was a source of carbon. I know this confusion on signs is difficult to handle, but I just wanted to raise awareness that it contributed to making this section difficult to follow. I apologize if I got these comparisons incorrect, but that indicates that more careful discussion and use of terminology would be helpful.

**Ocean Response**

I thought the second half of the discussion in this section was clear and informative; however, I had a two questions on Figure 6. First off, the caption states that it's "Integrated Primary Production," but doesn't define what that is. I assume that's the same as NPP? I'm unfamiliar with that terminology, so it might be worth clarification. Second, the values in the text and the figures don't match up. In the text the global mean ACCESS-ESM1 NPP is 46 PgC/yr, but from what I can tell this must be for the entire simulation. This is then compared to SeaWIFS value of 52 PgC/yr for 1998-2005. Upon reading this, I expect ACCESS-ESM1 to be lower than SeaWIFS in Figure 6; however, looking at it, ACCESS-ESM1 is higher than SeaWIFS for these years. I personally think it would be better if these comparisons were values representing the same time period, both to have a fair comparison and to match up the values with what is seen in Figure 6.

**GPP**

I would consider removing the first sentence in the second paragraph, as it is subjective and is not needed with the supporting text. I would then combine the first and second paragraphs to have a complete discussion. I would also remove or modify the final sentence in the second paragraph that states that containing nitrogen and phosphorous "ensures a more realistic simulation." While I

think there is evidence that including nitrogen and phosphorous is beneficial in many circumstances, that alone does not ensure a model outperforms one where these are not included but instead has more realistic representations of other important processes.

The second section in this section discussing the mean annual cycle of GPP is again quite subjective, and perhaps it wouldn't be too difficult to provide a few quantitative statements.

For the final discussion in the section on IAV, first off I was wondering how IAV is calculated? I can think of a couple of different methods, and it is not defined how they actually calculate the values that are shown. For this section, I suggest a PDF of errors, therefore when you say there is good agreement in the spatial pattern, it can be backed up with "x% of the globe has errors less than x kg C/m2." I will also note that the labels in Figure 9 were particularly hard to read.

**CNP Pool Sizes**

The discussion on the HWSD and soil carbon I found to be quite subjective, again focusing on the comparisons being "good" or "generally good."

For the nitrogen comparison, first off please clarify what the value reported, 85 Pg N, represents (i.e. global over entire simulation?) And just a thought: it might be interesting to show or state how this evolves in time, similar to the time-series shown for carbon.

For the phosphorus discussion, I would suggest either removing the "slightly" modifier used in the discussion on how the model results are lower than the estimated range, or give references on how this is smaller than previous modeling estimates to give a frame of reference for that adverb.

**Ocean Carbon**

The figures in this section are out of order. I would recommend swapping Figures 12 and 13, rather than discussing Figure 13 first and then going back to 12.

**Ocean NPP**

This section shows and discusses mean seasonal cycles; however, after discussing seasonal aspects such as shifts in timing, the section ends with the disclaimer that the timing cannot be compared. I'm wondering, if that's the case, might it be better to just look at amplitudes in a different format, such as a table or bar graph? If you leave the figures, I would recommend moving this statement to the beginning of the discussion and making it clear the timing aspects are only being compared between ACCESS-ESM1 and CMIP5. With the figure as it is currently, I again found the accompanying text subjective, which might also support looking into a table or bar graph format for this section to provide more quantitative comparisons.

I also want to note on Figure 13 the Southern Ocean has different values on the x-axis, which was not mentioned. This was confusing because at first it appears the Southern Ocean does not have a seasonal cycle, but from what I can tell (which is difficult given the large range), the amplitude is comparable to the other oceanic regions.

**Ocean Sea-Air CO2 Fluxes**

In this discussion, the authors state that the Southern subtropical gyres overestimate the observed sea-air flux; however, when I look at the figure, it looks to me the biggest uptake is occurring on the coastlines. I personally would like more of a mention of this, and maybe a discussion on why coastlines, particularly off the west coast of Australia, are taking up so much carbon compared to the broader ocean gyres.

In the second paragraph of this section, the second sentence beginning with "Furthermore it appears that globally…." doesn't make sense to me, as I don't understand what it is that lies outside the range. Usually regional analyses reveal why global results occur, and it looks like the Southern Hemisphere is the main contributor to the ACCESS-ESM1 global seasonal flux anomaly? The text does go on to state this, so you may just consider removing or modifying that sentence. Later in that same paragraph, the text states that the Northern Hemisphere has fluxes larger than observed but within the range of CMIP5; however, to me it looks like there are several months when ACCESS-ESM1 is outside the range (i.e. Jan, Feb, May). Since the same disclaimer was put on this paragraph (which again I would suggest moving to the beginning of the paragraph if these figures are kept), I would consider a different format, such as a table or bar graph, or even keeping the seasonal figures but not showing the seasonal cycle in the observations if you don't believe them, and instead using a solid line.

**Anthropogenic Inventory**
The discussion is again quite subjective. Also, in Figure 15, there is a reference to Key et al. (2004) that is confusing and not in the text (I'm unsure if the figure appears in both papers?) Upon finishing the paper, I see the reference in the conclusions, but you might want to somehow clarify or include it in the text during this discussion.

**Atmospheric CO2**
I think it would be better to combine the first paragraph with the next sentence following it (starting "Therefore, our…").

**Conclusions**
I think the conclusions would be stronger if some quantitative assessments were provided. I also noticed the order of the conclusions was not the same as the order presented in the text, but neither was it combined in a more succinct fashion to group conclusions. Simply reordering the paragraphs would make the text more consistent.

In the second-to-last paragraph, I was confused by the statement that "Seasonally the ACCESS-ESM1 appears biased toward the Southern Hemisphere." I'm not sure how to interpret this statement, as to me it's using both temporal and spatial references combined in an unclear fashion. This same type of statement occurs later in the sentence "Globally the annual mean is well captured but biased to low latitudes." I recommend reworking those sentences to clarify the intent of the text. Also, a quantification of how it is "well captured" would strengthen the statement.

**Technical Comments:**
Overall I have very few technical comments. My main technical comments are on comma usage. I personally find well-placed commas can aid in reading the text, but am not a comma expert, so feel free to take or leave the suggestions.

**Abstract**
Line 8: "reproduced; however, "
Line 9: "anthropogenic aerosols, "

**Introduction**
Line 19: "resolutions, and"
Line 21: remove comma "processes through"

Page 2, Line 7: "carbon cycle, an overestimation"
Page 2, Line 18: "soil carbon, especially in the northern latitudes, "
Page 2, Line 21: "However, "
Page 2, Line 28: "CMIP5, but in a"
Page 2, Line 31: "First, we briefly"

**Model Configuration, Simulations and Comparison Data**
Page 3, Line 12: "Wang et al., 2011), which"
Page 3, Line 14: "coupled, and"
Page 3, Line 16: "Consequently, global"
Page 4, Line 5: "negligible, and"
Page 4, Line 22: "simulations, we focused"
Page 4, Line 31: add a space "*precipitation*: Climate Research"

**ACCESS-ESM1 Climatology**
Page 6, Line 2: add an s "external forcings"
Page 7, Line 1: "lower albedo, especially"
Page 7, Line 2: "hemisphere, and consequently"
Page 7, Line 32: "Interestingly, "

**ACCESS-ESM1 Carbon Cycle Response**
Page 9, Line 16: check the text surrounding the reference, either putting the reference in
parenthesis or modifying the text

**Evaluation of the Prsent Day Carbon Cycle**
Page 13, Line 14: remove the s "encouraging result."
        (Just as a note, personally I would change that statement to something less subjective such
        as it falls within the observed range.)
Page 13, Line 26: "Table 2, and"
Page 14, Line 14: "(ACCESS-ESM1), respectively"
Page 14, Line 17: add a space "2 months."
Page 16, Line 5: change from is "fluxes are not well captured"
Page 16, Line 23: "well reproduced, with"
Page 16, Line 7: "As observed, the amplitude"

**Conclusions**
Page 16, Line 22: remove comma "hemisphere and underestimates"
Page 16, Line 29: remove by "is about 20%"
Page 16, Line 31: "globally, therefore increasing NEE."
Page 17, Line 4: "into tracers, which"

---

## Author Comment (AC1) · 24 Aug 2016

We would like to thank the referee for the review of this manuscript and their constructive comments. Our response to each comment is below with the referee's comments highlighted in bold typeface. Where appropriate, we have also included relevant changes in the revised manuscript in italic typeface.

**My main concern with this model is the length of time taken to achieve a spun-up state. Law et al document this nicely, but I think it requires more discussion in this paper too what the implications are. The drifts in carbon stores are still non-egligible even after 800 years of spin-up (your start point here). I think this should be laid out explicitly before the analysis starts. You do, in the land carbon section, acknowledge this and subtract the control run drift. But unless a reader has been through Law et al they would not know how big this drift is. For the ocean it is more important still, and the ocean section does not mention this at all. The drift of circa 0.7 GtC/yr (figure 11a in Law et al) is of similar magnitude to your historical fluxes (I assume these are corrected for the drift). If a reader hadn't seen that figure then they would not realise from this paper the size of drift being subtracted.**

We will include a paragraph in the Simulations section 2.2 to acknowledge the drift in carbon stores more explicitly for land and ocean. We will also make it clear in the ocean section that the historical fluxes have been corrected for the drift.

*As noted in Law et al. (2015) the net carbon fluxes for land and ocean did not equilibrate to zero. At the end of the control run (i.e. year 800 to 955), global NEE is 0.3 PgC/yr for PresLAI and 0.08 PgC for ProgLAI. The net autgassing from the ocean is about 0.6 PgC/yr at the end of the control run. We take this drift into account when we calculate the net uptake of carbon for land and ocean.*

**In CONCENTRATION-driven runs like this, you can of course force the correct CO2 and correct for the drift after the simulation. But that is not possible in an EMISSION-driven run, and such a drift would cause a massive drift in atmospheric CO2 render-ing an emissions-driven historical run meaningless. This would, at present preclude use of this model in C4MIP for example which would be a great shame. The latest C4MIP protocol (Jones et al 2016, GMDD - CMIP special issue) recommends a maximum acceptable drift of 10 GtC per century. I therefore throughly recommend that ACCESS modellers attempt to find accelerated means to derive a spun-up state in time for CMIP6. There are numerous options, such as running offline (for either land or ocean), or using reduced turn-over time techniques as per Koven et al for CLM (http://www.biogeosciences.net/10/7109/2013/bg-10-7109-2013.pdf). as a final word on this, lack of carbon conservation would also be more of an issue for E-driven runs.**

We are currently exploring various spin-up options (including offline simulations) to improve the drift for ocean and land. If successful, this will then be implemented in future versions of ACCESS-ESM.

**My second concern is the lack of land-use change as a forcing. You already know this, and acknowledge it in the paper, so no revisions to the manuscript are required, but I just take this opportunity to stress that simulations of contemporary and future climate/carbon cycle are very much reduced in usefulness if they lack the**

**very large land-use forcing of the land carbon cycle. Implementing this for CMIP6 is also therefore a priority I would say.**

We agree with the referee and it is a high priority for us to include land use change in future versions of ACCESS-ESM.

**Having quickly read Law et al before I reviewed this one I was struck that there was not an evaluation there of (land) carbon stocks. I do feel that the land carbon modelling community have become fixated on evaluation against fluxes to the detriment of stocks and residence times. This is beginning to change and I was pleased to see some discussion of carbon stocks in this paper. It would be nice to see the time changes in these though as well - could table 2 be extended to show pre-industrial and present day stocks? In the discussion on biomass you mention that your results are higher than observational based estimates - but of course you lack land-use change as a driver. So it could easily be expected that your biomass is of the order 100-150 GtC too high due to this. If you masked for present day agricultural regions you would probably get a much closer fit to expected global totals. So your simulation is actually not bad.**

In a revised version we will extend Table 2 to show both, pre-industrial and present day stocks. We thank the referee for the comment about the size of our biomass pool and we will include this statement in a revised version.

| Pool | Pre-industrial PresLAI C | N | P | ProgLAI C | N | P | Present day PresLAI C | N | P | ProgLAI C | N | P | Historical change C PresLAI ΔC | ProgLAI ΔC |
|---|---|---|---|---|---|---|---|---|---|---|---|---|---|---|
| Biomass | 611 | 5.7 | 0.31 | 731 | 6.15 | 0.33 | 670 | 6.2 | 0.34 | 807 | 6.8 | 0.37 | 69.5 | 87.2 |
| Litter | 117 | 0.85 | 0.04 | 149 | 1.02 | 0.05 | 126 | 0.9 | 0.05 | 163 | 1.1 | 0.06 | 7.6 | 12.3 |
| SOC | 1034 | 82 | 9.6 | 1187 | 86.1 | 11.9 | 1050 | 83.4 | 10.1 | 1217 | 88.5 | 12.6 | 20.5 | 37 |
| $\sum$ | 1762 | 88.6 | 10.0 | 2067 | 93.3 | 12.3 | 1846 | 90.5 | 10.5 | 2187 | 96.4 | 13.0 | 97.6 | 136.5 |

**Abstract - you can say that aerosol forcing is large or larger than other models. But don't say "over-sensitive" as we simply don't know. Maybe this is correct ...**

We will change this statement as suggested in a revised version.

**section 2.3 - I like the comparison vs CMIP5 models. This is a nice way to put model-data discrepancies in context. But why do you only include 5 CMIP5 models (counting the 2 IPSLs as one model). Anav et al used more than twice that - any reason not to use the full set?**

We agree that ideally we would have been able to use more Earth System Model (ESM) results like Anav et al., though unfortunately, at the time we were doing the analysis not all the output was readily available from the ESG servers. After careful examination of our final results and those of Anav et al., we are confident that our figures still capture the spread of CMIP5-ESM results and more models may only serve to broaden the spread of the models.

**in a couple of places (e.g. start sec 3.1) you mention the variability of the land**

**sink and/or the atmospheric CO2. You could go further and use this as an evaluation metric. Both Cadule et al (2010, GBC) and Cox et al (2013, Nature) show the power of the C-cycle sensitivity to ENSO on inter-annual timescales as a really strong evaluation metric and constraint on the sensitivity of ESMs**

We thank the referee for his suggestion and will consider this in future evaluation papers using the next version of ACCESS-ESM.

**p.7, line 21. I don't understand why you attribute your slower warming to the initial warm bias - how do you know the warm bias causes this? Could just be under-sensitive SSTs not related to a bias.**

The discussion was intended to be descriptive rather than mechanistic.  We clarify now the text so as not to attribute the reduced warming to the warm bias early in the historical period.

**p.8 line 3. I was amused to see that having an error different from other models was "encouraging"! not sure why! Can you say why it is better to have the opposite MLD bias from other models?**

Encouraging in the sense that simulating well winter mixed layer depths are critical for setting interior ocean properties  supplying nutrients to the upper ocean to fuel the biologically active growing season. We have now augmented the manuscript to say:

*In the higher latitude the winter mixed layers are well captured by ACCESS-ESM1 Figure 4. This is encouraging given that many ocean models tend to underestimate winter mixed layer depths (Sallee et al (2013) and Downes (2015)). Simulating winter mixed layers correctly is critical for setting interior ocean properties supplying nutrients to the upper ocean to fuel the biologically active growing season (Rodgers et al, 2014).  However, in contrast to the winter, ACCESS-ESM1 appears to systematically underestimate mixed layer depths in the high latitude ocean in summer, ~60% (or 30-40m) in the Southern, Pacific and Atlantic Oceans. In the Southern Ocean, in particular, the underestimation of summer mixed layer depths is consistent with Sallee et al (2013) and Haung et al (2013) who showed that most CMIP5 models underestimate summer mixed layer depths. Haung et al (2013) attributed this to a lack of vertical mixing in CMIP5 rather than sea surface forcing related to individual models, this is consistent with Downes et al (2015) who showed that these biases are also present in the ocean only simulations of ACCESS-ESM1 (ACCESS-O).*

**in general I thought this MLD section (3.2) was a bit superficial - can you be a bit more quantitative in your comparison and description? the figure shows the data but it can be hard to tell from there if the differences you describe are of the order of a few % or 10s of % or factors of 2 or more or what? And can you mention your confidence in the ons? presumably global maps of MLD are not directly observed but must have certain extrapolation uncertainties and so on. Are some areas/seasons of the oceans better sampled than others etc …**

To better illustrate the chages we have added a lower panel to Figure 4 that is the percentage changes of the difference between the (ACCESS-ESM1 – obs/obs) *100. This illustrates the relative changes as suggested by the reviewer.  Also please see above for an enhanced discussion.

Regading the number of samples and coverage of the obs we have added the following paragraph:

*Ocean mixed layer depths are compared with the observations following DeBoyer Montegut et al (2004) based on more than 880,000 depth profiles from research ships and ARGO profiles, and based on a 0.03kg/m³ density change from the surface. Significant advances in autonomous measurement platforms have allowed the mixed layer to be increasingly well constrained in all seasons across the global ocean.*

[Figure]

**p. 8 line 21. Can you define what you mean by IAV. Interannual Variability I know, but how do you turn this into a number? is it the standard deviation of a time series of annual means? in which case is the time series de-trended first? Etc**

The interannual variability (IAV) is calculated as the standard deviation for the de-trended annual mean values. This explanation will be included in a revised version.

**p.9 When discussing historical changes in land carbon can you split into veg and soil changes (e.g. put in table 2). You could compare directly with the 2 models in Jones et al (2013) which also don't have land-use forcing (dashed lines in figure 2). You could probably also compare with model results from detection-and-attribution studies which ran with/without certain forcings. The are probably various no-land-use runs to look at.**

We will included the change in land carbon for vegetation, litter and soil for both scenarios (prescribed LAI and prognostic LAI) in Table 2 and also compare these values against simulation of the two models without land use change from Jones et al. (2013) in a revised version.

| | Pre-industrial | | | | | | Present day | | | | | | Historical change C | |
| | PresLAI | | | ProgLAI | | | PresLAI | | | ProgLAI | | | PresLAI | ProgLAI |
| Pool | C | N | P | C | N | P | C | N | P | C | N | P | $\Delta C$ | $\Delta C$ |
|---|---|---|---|---|---|---|---|---|---|---|---|---|---|---|
| Biomass | 611 | 5.7 | 0.31 | 731 | 6.15 | 0.33 | 670 | 6.2 | 0.34 | 807 | 6.8 | 0.37 | 69.5 | 87.2 |
| Litter | 117 | 0.85 | 0.04 | 149 | 1.02 | 0.05 | 126 | 0.9 | 0.05 | 163 | 1.1 | 0.06 | 7.6 | 12.3 |
| SOC | 1034 | 82 | 9.6 | 1187 | 86.1 | 11.9 | 1050 | 83.4 | 10.1 | 1217 | 88.5 | 12.6 | 20.5 | 37 |
| $\sum$ | 1762 | 88.6 | 10.0 | 2067 | 93.3 | 12.3 | 1846 | 90.5 | 10.5 | 2187 | 96.4 | 13.0 | 97.6 | 136.5 |

When we calculated the change in land carbon based on the pool sizes rather than using the net flux, we noticed that the total land carbon uptake over the historical period is actually much smaller than stated in the paper: 98 PgC instead of 128 PgC (PresLAI) and 137 PgC instead of 154 PgC. This is because we used an earlier section of the control run (years 325-480) to calculate the drift. However, the historical runs we describe in the paper were started at year 800 from the control run and this section (i.e. year 801-955) shows a much smaller drift for both scenarios. Using the correct drift applied to the net flux over the historical period provides now about the same total land carbon uptake as calculated via the pool sizes. The numbers for the total uptake of carbon by the land will be corrected in a revised version.

We have also updated the NEE time series plot (Fig. 5c) using the correct drift, although changes are hardly visible, particularly for the 5yr running mean.

**Sec 4.2. Despite the large drift your historical ocean sink does look a very close match to the obs. Can you also quote a cumulative uptake here?**

We have added the following text to the paper:

*The cumulative uptake of carbon by air-sea $CO_2$ fluxes in the period 1959-2005 from ACCESS-ESM1 is 83 PgC which is good agreement with the GCP value of 82PgC over the same period.*

**sec 5.1.2. I was curious to see that your prescribed LAI didn't match your obser-**

**vations. Can you explain why not? You illy that this is because there are differences between observed datasets - which is of course true. But then why do you choose one dataset to prescribe LAI to the model, and then a different one to evaluate against? If one is better than the other can you use it for both?**

The prescribed LAI we use in ACCESS-ESM1 is based on MODIS observations and has no interannual variability. We decided to compare this with a more recent LAI product (MODIS/AVHRR combination) to (a) investigate if there are significant differences in the mean seasonal cycle for present day (as shown in Fig. 10) and (b) to investigate what the interannual variability in the observations looks like (not show in the manuscript).

We don't think that one LAI product is better than another. Historically, CABLE uses a prescribed LAI with no interannual variability. However, in future CABLE/ACCESS versions we might update our prescribed LAI to a product with interannual variability.

As mentioned in the paper, LAI products differ from each other because different sensors and algorithms are used. In addition, the LAI products shown here are also based on different observing time periods.

**p.13 discussion of different carbon stocks for the two configurations. Given both have very similar GPP, the large difference in biomass is presumably due to residence times? Having GPP further north in compared to the tropics means that for the same global GPP you have a higher biomass? You might consider next time an evaluation of turnover times - e.g. as per Carvalhais et al (2014, Nature)**

We agree with the referee, that the difference in carbon stocks for the two scenarios (prescribed LAI and prognostic LAI) can probably be explained by a difference in residence time. We thank the referee for pointing this out and we aim to include an evaluation of turnover times in future work.

**section 5.2.1 / 5.2.2. Can you swap the order of figures 12 and 13? you describe 13 first.**

The order of these figures will be changed in a revised version.

**sec 5.2.3. You compare global totals between model and GLODAP, but the map shows missing areas in the ons dataset. So should you first mask out the model to match the same area before you compare totals? (or quote a full AND a masked number for the model). The values in the Arctic for example are very high but missing in the obs (although of course the area is smaller than this projection makes it look).**

We have redrawn the ACCESS-ESM1 map with only ares that observations exist. Furthermore we now quote the number from comparison with GLODAP and for the entire domain:

*The inventory for the period 1850-1994 in ACCESS-ESM1 is 132 PgC, which is close to the estimated value from GLODAP of 118 +/- 19PgC (Sabine et al, 2004) over the same domain. This suggests that despite a somewhat limited representation of the seasonal cycle of sea-air $CO_2$ fluxes in key regions of anthropogenic uptake such as the Southern Ocean, ACCESS-ESM1 is doing a very good job, spatially and temporally, of capturing*

*and storing anthropogenic carbon. If the entire domain (including the Arctic Ocean) is integrated the anthropogenic uptake is 143 PgC over the same period.*

[Figure]

ACCESS Vertical Anthropogenic Inventory                GLODAP Vertical Anthropogenic Inventory

**sec 5.3. It seems reasonable that the land drives most of the seasonal cycle at your ons stations except the south pole. I like the way you have labelled the land and ocean CO2 separately to be able to diagnose this. But I couldn't work out why the contribution of the ocean at the south pole looks different for your two LAI configurations. If I read the figure correctly then the role of the ocean is given by the blue solid minus the blue dashed line and the red solid minus the red dashed line. These are quite different by eye - e.g. the blue lines are quite far apart in December and the red lines quite close. So why does your LAI treatment affect your December ocean fluxes so much?**

The LAI scenario should not affect the ocean carbon fluxes and we thank the referee for pointing out this inconsistency. By mistake, we showed the total flux using two different ocean configurations in Figure 16 (mean seasonal cycle ot atmospheric CO2). Using the same ocean configuration for both scenarios (prescribed LAI and prognostic LAI) solves this issue and the dashed lines (land carbon flux) and solid lines (total flux) are much closer together (see figure below). Figure 16 will be updated in a revised version accordingly.

[Figure]

**Table 2. Can you add a final row at the bottom of "total"**

We will do this in a revised version.

| Pool | Pre-industrial | | | | | | Present day | | | | | | Historical change C | |
| | PresLAI | | | ProgLAI | | | PresLAI | | | ProgLAI | | | PresLAI | ProgLAI |
| | C | N | P | C | N | P | C | N | P | C | N | P | ΔC | ΔC |
|---|---|---|---|---|---|---|---|---|---|---|---|---|---|---|
| Biomass | 611 | 5.7 | 0.31 | 731 | 6.15 | 0.33 | 670 | 6.2 | 0.34 | 807 | 6.8 | 0.37 | 69.5 | 87.2 |
| Litter | 117 | 0.85 | 0.04 | 149 | 1.02 | 0.05 | 126 | 0.9 | 0.05 | 163 | 1.1 | 0.06 | 7.6 | 12.3 |
| SOC | 1034 | 82 | 9.6 | 1187 | 86.1 | 11.9 | 1050 | 83.4 | 10.1 | 1217 | 88.5 | 12.6 | 20.5 | 37 |
| Σ | 1762 | 88.6 | 10.0 | 2067 | 93.3 | 12.3 | 1846 | 90.5 | 10.5 | 2187 | 96.4 | 13.0 | 97.6 | 136.5 |

**general comment on figures - maybe a personal preference but please can you add a legend so that I can easily read which line is which. The text in the captions is very good - but I have to read a whole paragraph to spot what the red/blue lines are.**

**Would be great to simply see a legend with this in as well as the detail in the caption text.**

A legend will be added to all relevant figures in a revised version.

---

## Author Comment (AC2) · 24 Aug 2016

Response to Referee RC2:

We would like to thank the referee for the review of this manuscript and their constructive comments. Our response to each comment is below with the referee's comments highlighted in bold typeface. Where appropriate, we have also included relevant changes in the revised manuscript in italic typeface.

**The main focus of these specific comments is to highlight text that could benefit from being more quantitative rather than subjective. In addition, there are some specific suggestions for clarifications aimed at improving the overall flow and clarity of the manuscript. Before starting the specific suggestions, one other general remark I have is that the figure axes and color bar labels are very small and hard to read.**

We will improve the readability (i.e. increase size of labels) of all figures in a revised version.

**Abstract**
**The abstract provides an overall view of how well ACCESS-ESM1 performs; however, they provide this information using terms such as "good" and "performs well". How well? Good according to whom? Is your good the same as my good? First off, the authors state ACCESS-ESM1 overestimates the seasonal amplitude of LAI, but do not attach any quantities to this statement. Rather than providing a quantitative assessment of this statement, the reader is left to wonder if this is a substantial bias, perhaps even being prohibitive of using this model in the future, or merely a relatively minor difference that is offset by other positive features. The paper then continues with the statement that the oceanic and land fluxes "show good agreement with the observations", but again, no metric is used. How big are the differences, what is the error, or how closely correlated, both spatially and temporally, are they to observations and/or other CMIP-5 models? In the last overview comparison, the authors state that the seasonal cycle is "close to the observed seasonal cycle", but as the reader I have to wonder how close is close? I believe that putting quantifiable metrics on at least some of these statements will strengthen the concluding remark that ACCESS-ESM1 is indeed a useful tool.**

We will include more quantitative statements in the abstract in a revised version.

**Observations**
**The model evaluation (later in the text) proceeds through a straightforward succession of comparisons; however, the order of these comparisons is different than the order the data is listed in. For clarity and consistency, it would be an easy fix to reorganize this section to present the data in the order that they are used.**

We will reorganise section 2.4 (Observations) in a revised version, so that data are presented in the order they are used later in the comparison.

**Land Temperature and Precipitation**
**This section presents a time-series comparison of temperature and precipitation, but contains very little quantitative statements. For example, the temperature anomalies are "close to the observed anomalies through most of the period". Again I have to wonder, how close is close? Perhaps a difference plot showing the errors would be useful? Or perhaps a correlation coefficient that may or may not be**

**significant? Some sort of metric on this statement would be much more enlightening to the reader. For example, as I look at Fig. 1 myself, I see the author's point that ACCESS-ESM1 is lower than the observations 1965-2005; however, to my eye I also see the model looks quite a bit lower in the 1940s. I think that a difference plot would help identify these areas, rather then relying on the reader to have to assess the differences between the models and the observations on eyesight alone. Another example later in this section occurs in the precipitation anomaly discussion: the authors state the differences are "generally small," but provide no values to suggest what is meant by small. This is followed by the statement that "the simulations compare well with observed rainfall anomalies until about 1950," with no supporting metric such as minimal differences or significant correlation coefficients to back up this statement. Overall, I found that this entire section had very few quantitative comparisons, instead relying heavily on subjective terminology, making me believe that at least some measurable metrics would improve the paper.**

We have attached an updated Fig. 1 showing differences for temperature and precipitation anomaly (simulated – observed).  However, we might not include the difference plot in the manuscript, because we don't think it adds a lot more information. Instead, we will focus on decadel mean differences and refer to the values in the text as appropriate:

*Both ACCESS-ESM1 simulation scenarios (PresLAI and ProgLAI) show similar temperature anomalies over most of the historical period, being close to the observed anomalies through most of the period (decadal mean difference smaller than 0.2 K), apart from the 1940s where the PresLAI scenario shows a larger negative anomaly (decadal mean difference of about 0.37 K), which will be discussed later. From about 1965-2005 anomalies are by up to 0.4 K (decadal mean difference) lower than observations for both scenarios.*

*ACCESS-ESM1 simulations compare well with observed rainfall anomalies until about 1960 (decadal mean difference smaller than 8 mm/yr), with the exeption of the period 1911-1920 for PresLAI (decadal mean difference of about 12 mm/yr) and the period 1951-1960 for ProgLAI (decadal mean difference of about 17 mm/yr). After that, observed anomalies are mostly higher than the simulation results (decadal mean difference of up to 41 mm/yr), a feature also seen in the ACCESS1.3 historical ensemble (Fig. 6a, Lewis and Karoly 2014).*

The anomaly in the 1940s is already discussed in the manuscript on page 6, lines 26-30.

We do not think that a correlation coefficient would be a very meaningful metric to assess the errors between simulated and observed temperature and precipitation. According to Anav et al. (2013), there is no reason to expect models and observations to agree on the phasing of internal interannual variations. We therefore calculate the model variability index (MVI) to analyse the performance of ACCESS-ESM1 for temperature and precipitation. The MVI compares the models variability at every grid cell, which is then averaged for the globe.  Perfect model – observation agreement would result in an MVI value of 0. For example, for temperature we calculate an MVI of 0.3 (PresLAI) and for precipitation an MVI of 1.7 (PresLAI) over the period 1901-2005. We will include this information in a revised version of the paper:

*The interannual variability in temperature is well reproduced by both ACCESS-ESM1 scenarios, showing an MVI of 0.3 (PresLAI) and 0.4 (ProgLAI) for the period 1901-2005.*

*According to Anav et al. (2013) only a few CMIP5 models show an MVI of lower than 0.5 (although their calculation is based on present day, i.e. 1986-2005).*

*For precipitation we calculate an MVI of 1.7 (PresLAI) and 1.8 (ProgLAI) for the period 1901-2005, which suggests that the IAV is not well represented in ACCESS-ESM1. However, according to Anav et al. (2013) none of the CMIP5 models had an MVI close to the threshold of 0.5. Also note that for the calculation of the MVI for precipitation we had to exclude 60 land points (mainly coastal points) due to inconsistencies in the regridding.*

Anav, A., Friedlingstein, P., Kidston, M., Bopp, L., Ciais, P., Cox, P., Jones, C., Jung, M., Myneni, R., and Zhu, Z.: Evaluating the Land and
Ocean Components of the Global Carbon Cycle in the CMIP5 Earth System Models, J. Climate, 26, 6801–6843, 2013.

[Figure]

**The last paragraph in this section discusses the timing of precipitation anomalies versus volcanic eruptions. The authors point out a reduction in precipitation following eruptions, with the one exception of El Chichon; however, when I looked at the time-series, I did not see a decrease after the Santa Maria eruption in addition to El Chichon. In fact, looking at the time-series, the decrease after volcanic eruptions did not stand out to me, especially when the authors note the reduction following the Mt. Pinatubo eruption, which does stand out, is too far away from the eruption date to be related. I think a more quantifiable analysis of the magnitude and the timing of this decrease, in days or weeks or some stated time-scale, would be helpful.**

We agree with the referee that there is no decrease in precipitation visible following the eruption of Santa Maria. We will include this statement in a revised version of the manuscript.

However, we did not say that a reduction in rainfall following Mt. Pinatubo is too far away from the eruption date. In fact, along with Krakatao and Mt. Agung, the Mt Pinatubo event shows a significant reduction in precipitation anomalies immediately after the eruption.

Sea Surface Temperature and Mixed Layer Depth
**The second full paragraph of this section discusses spatial patterns of sea surface temperatures, except no time period for the analysis is provided, not even in the caption for Figure 3. Are these differences over the entire simulation or a selected time period? Shifting to the text, the authors state that ACCESS-ESM1 "produces very heterogeneous differences from observations." Reading this, it was unclear to me what was meant. When I turned to look at the figure I expected to see random errors; however, in my opinion the differences are spatially coherent in latitude bands. Then in this same discussion they state "there do not appear to be strong seasonal biases," but with this terminology I have to wonder are there or aren't there? Then they state the exception of the North Atlantic, which has a coherent bias towards cooler temperatures, but to me it looks like this is a year-round bias more than a seasonal bias. It perhaps does vary in magnitude with season, being a larger bias in August, but it looks like the sea surface temperature here is underestimated year round. Further looking at Figure 3, I also see a flip-flop in errors in the southeastern Pacific Ocean, with positive differences in August and negative differences in February. To me, having differences that vary with time of year makes it seasonal, but this region is not mentioned. I think this section could benefit from more careful wording and analysis.**

We apologies for the oversight and have now stated in the caption and text that the time period of the figures is the IPCC historical period 1986-2005. We have rewritten the text to better convey our intended meaning and analysis. Indeed the spatial patterns of the warming are perplexing better upon a much closer examination appear to be associated with the Met Office Unified Model (MetUM) which has known biases and which ACCESS-ESM1 utilizes as it atmospheric model, indeed similar biases are seen in HadGEM2 that also use the MetUM.

**The ocean mixed layer depth discussion could also benefit from a more quantitative analysis, rather than using statements like "appears to slightly overestimate the**

**depth in winter" and "appears to underestimate the depths in summer," with no statistical support for these subjective comments. Also, in Figure 4 in this section, the caption states that differences are shown, suggesting a difference plot; however, the figures just show the results from both the model and the observations. I would suggest rewording the caption to avoid confusion.**

To address the reviewer's comments we have rewritten the section on mixed layer depth comparison, and added an additional figure showing the percentage changes in mixed layer depth between the observations and ACCESS-ESM1. This allows the changes between the obs and model to be quantified relative to the total observed mixed layer depth.  We have also updated the caption to reflect the reviewers concerns.

*In the higher latitude that the winter mixed layers are well captured by ACCESS-ESM1 Figure 4. This is encouraging given that many ocean models tend to underestimate winter mixed layer depths (Sallee et al (2013) and Downes (2015)). Simulating winter mixed layers correctly is critical for setting interior ocean properties supplying nutrients to the upper ocean to fuel the biologically active growing season (Rodgers et al, 2014).  However in contrast to the winter, ACCESS-ESM1 appears to systematically underestimate mixed layer depths in the high latitude ocean in summer, ~60% (or 30-40m) in the Southern, Pacific and Atlantic Oceans. In the Southern Ocean, in particular, the underestimation of summer mixed layer depths is consistent with Sallee et al (2013) and Haung et al (2013) who showed that most CMIP5 models underestimate summer mixed layer depths. Haung et al (2013) attributed this to a lack of vertical mixing in CMIP5 rather than sea surface forcing related to individual models, this is consistent with Downes et al (2015) who showed that these biases are also present in the ocean only simulations of ACCESS-ESM1 (ACCESS-O).*

[Figure]

a) ACCESS-ESM1 Feburary MLD

b) ACCESS-ESM1 August MLD

c) Observed Feburary MLD

d) Observed August MLD

Feburary MLD: (ACCESS-ESM1 -Obs)/Obs

August MLD : (ACCESS-ESM1 -Obs)/Obs

**Land Carbon Response**

I think it would be helpful to include the MODIS/AVHRR LAI data in Figure 5b. I realize this would only be for the last few years of the simulation, but the benefit is that it would provide a reference to the simulated LAI values. After reading it through, I discovered there is a section on LAI where it is discussed in more detail, but this figure comes first, and when I read it I was wondering how it compared.

Observation based LAI data (MODIS/AVHRR) have been included in Fig. 5b for the period 1982 to 2005. For comparison we have also included the prescribed LAI used in ACCESS-ESM1, which has no interannual variability. The updated figure will be included in a revised version of the paper

[Figure]

**I found the discussion in the last two paragraphs of this section confusing. I loved to see values and uncertainties; however, it was unclear to me what values were comparable. The section starts with a discussion on carbon uptake, which at first I assumed represents total uptake, or GPP. But from what I could tell, the same values provided for ACCESS-ESM1 were then used in the following paragraph, which talks about NEP. I read the section several times, but only found the one value of 154 Pg C for prognostic ACCESS-ESM1. I was then confused when it was stated that this value is "at the low end of the CMIP5 range," when that range is estimated to be from -59 to 18 PgC according to Shao et al. (when outliers are not included) or from**

**-124 to 50 PgC from Jones et al. Based on these numbers, doesn't ACCESS-ESM1 take up and store much more carbon (+154 Pg C) than the CMIP5 models? It doesn't help that the signs make this analysis even more confusing. Since NEP and uptake were discussed, I assumed a negative value was a source of carbon. I know this confusion on signs is difficult to handle, but I just wanted to raise awareness that it contributed to making this section difficult to follow. I apologize if I got these comparisons incorrect, but that indicates that more careful discussion and use of terminology would be helpful.**

The section "Land carbon response" discusses the impact of the historical forcing on some carbon related variables, i.e. gross primary production (GPP), leaf area index (LAI) and net ecosystem exchange (NEE) and their interactions.

We actually do not discuss absolute values for GPP in this section, we focus mainly on interannual variability (IAV) and trend. Absolute values of GPP (i.e. mean GPP for present day) are discussed and compared against observations in section 5.1.1. "GPP".

The last two paragraphs analyse the total land carbon uptake over the historical period, which is the sum of the net ecosystem production NEP (opposite sign to NEE, i.e. NEP = -1 x NEE) from 1850 to 2005. Throughout the paper we consistently analyse the flux to the atmosphere (i.e. land to air and sea to air) which is commonly used for analysing CMIP5 modelling results. However, in order to calculate the uptake by land and ocean we need to reverse the sign.  We will clarify this in a revised version of the paper.

The value of 154 PgC represents the total land carbon uptake over the historical period for the scenario with prognostic LAI (i.e. cumulative NEP). Note, this value will be corrected in a revised version (see also reply to reviewer 1).

We currently do not consider disturbances such as land use and land cover change (LULCC) in our simulations, which means that our land carbon uptake is simply calculated based on NEP. The majority of CMIP5 models include LULCC in some form or the other, which makes it difficult to compare our calculated uptake against land carbon uptake from CMIP5 models. We tried to do this in two ways:

(a) we compare our results against cumulative NEP with values reported in Shao et al. (2013) with NEP ranging from 24 to 1730 PgC, which means ACCESS-ESM1 is at the lower end of this range.

(b) we compare our results against observational based estimates of land use emissions which are thought to be 108-188 PgC for the historical period. This means we get an almost neutral behaviour by accounting for LULCC in this way. CMIP5 models that include disturbances also estimate a neutral behaviour by providing an estimate of land carbon uptake of -59 to 18 PgC (Shao et al., 2013) and -124 to 50 PgC (Jones et al., 2013).

We will revise the whole section accordingly.

**Ocean Response**
**I thought the second half of the discussion in this section was clear and informative; however, I had a two questions on Figure 6. First off, the caption states that it's "Integrated Primary Production," but doesn't define what that is. I assume that's the same as NPP? I'm unfamiliar with that terminology, so it might be worth**

**clarification. Second, the values in the text and the figures don't match up. In the text the global mean ACCESS-ESM1 NPP is 46 PgC/yr, but from what I can tell this must be for the entire simulation. This is then compared to SeaWIFS value of 52 PgC/yr for 1998- 2005. Upon reading this, I expect ACCESS- ESM1 to be lower than SeaWIFS in Figure 6; however, looking at it, ACCESS-ESM1 is higher than SeaWIFS for these years. I personally think it would be better if these comparisons were values representing the same time period, both to have a fair comparison and to match up the values with what is seen in Figure 6.**

We apologise for any inconsistency, it should have been 51 PgC/yr, and we have now clarified the text to replace integrated primary production with net primary production and added a section explaining how NPP is calculated.

**GPP**
**I would consider removing the first sentence in the second paragraph, as it is subjective and is not needed with the supporting text. I would then combine the first and second paragraphs to have a complete discussion. I would also remove or modify the final sentence in the second paragraph that states that containing nitrogen and phosphorous "ensures a more realistic simulation." While I think there is evidence that including nitrogen and phosphorous is beneficial in many circumstances, that alone does not ensure a model outperforms one where these are not included but instead has more realistic representations of other important processes.**

As suggested by the reviewer we will remove the first sentence of the second paragraph and combine the first two paragraphs in a revised version. We will change the final sentence of the second paragraph to:

*ACCESS-ESM1 contains both nitrogen and phosphorus limitation, which may provide a more realistic simulation of carbon cycle uptake by the terrestrial biosphere.*

**The second section in this section discussing the mean annual cycle of GPP is again quite subjective, and perhaps it wouldn't be too difficult to provide a few quantitative statements.**

We will include more quantitative statements in the discussion of the mean annual cycle of GPP in a revised version.

**For the final discussion in the section on IAV, first off I was wondering how IAV is calculated? I can think of a couple of different methods, and it is not defined how they actually calculate the values that are shown. For this section, I suggest a PDF of errors, therefore when you say there is good agreement in the spatial pattern, it can be backed up with "x% of the globe has errors less than x kg C/m2." I will also note that the labels in Figure 9 were particularly hard to read.**

The interannual variability (IAV) is calculated as the standard deviation for the de-trended annual mean values. This explanation will be included in a revised version.

As suggested by the reviewer we have calculated the absolute error for present day mean GPP for each land grid point. For example, 95% of all land points have errors smaller than 0.5 kgC/m2/yr for the scenario with prescribed LAI (86% for the scenario with prognostic LAI). We will include those numbers in a revised version.

We will increase the size of labels in Fig. 9 in a revised version.

**CNP Pool Sizes**
**The discussion on the HWSD and soil carbon I found to be quite subjective, again focusing on the comparisons being "good" or "generally good."**

**For the nitrogen comparison, first off please clarify what the value reported, 85 Pg N, represents (i.e. global over entire simulation?) And just a thought: it might be interesting to show or state how this evolves in time, similar to the time-series shown for carbon.**

The 85 PgN represent the mean soil organic (SOC) pool size for the last 20 years of the historical period (1886-2005). In the manuscript on page 13, line 2 we stated that all pool sizes are calculated over the last 20 years of the historical period.

We will include initial pool sizes (i.e. spun up pools from pre-industrial simulation) for CNP for both scenarios in a revised version in Table 2 so that they can be compared against present day pool sizes:

| | Pre-industrial | | | | | | Present day | | | | | | Historical change C | |
| | PresLAI | | | ProgLAI | | | PresLAI | | | ProgLAI | | | PresLAI | ProgLAI |
| Pool | C | N | P | C | N | P | C | N | P | C | N | P | $\Delta$C | $\Delta$C |
| --- | --- | --- | --- | --- | --- | --- | --- | --- | --- | --- | --- | --- | --- | --- |
| Biomass | 611 | 5.7 | 0.31 | 731 | 6.15 | 0.33 | 670 | 6.2 | 0.34 | 807 | 6.8 | 0.37 | 69.5 | 87.2 |
| Litter | 117 | 0.85 | 0.04 | 149 | 1.02 | 0.05 | 126 | 0.9 | 0.05 | 163 | 1.1 | 0.06 | 7.6 | 12.3 |
| SOC | 1034 | 82 | 9.6 | 1187 | 86.1 | 11.9 | 1050 | 83.4 | 10.1 | 1217 | 88.5 | 12.6 | 20.5 | 37 |
| $\sum$ | 1762 | 88.6 | 10.0 | 2067 | 93.3 | 12.3 | 1846 | 90.5 | 10.5 | 2187 | 96.4 | 13.0 | 97.6 | 136.5 |

**For the phosphorus discussion, I would suggest either removing the "slightly" modifier used in the discussion on how the model results are lower than the estimated range, or give references on how this is smaller than previous modeling estimates to give a frame of reference for that adverb.**

We will remove the word "slightly" as suggested by the reviewer in a revised version.

**Ocean Carbon**
**The figures in this section are out of order. I would recommend swapping Figures 12 and 13, rather than discussing Figure 13 first and then going back to 12.**

We have now swapped the figures consistent with the suggestion of the reviewer.

**Ocean NPP**
**This section shows and discusses mean seasonal cycles; however, after discussing seasonal aspects such as shifts in timing, the section ends with the disclaimer that the timing cannot be compared. I'm wondering, if that's the case, might it be better to just look at amplitudes in a different format, such as a table or bar graph? If you leave the figures, I would recommend moving this statement to the beginning of the**

**discussion and making it clear the timing aspects are only being compared between ACCESS- ESM1 and CMIP5. With the figure as it is currently, I again found the accompanying text subjective, which might also support looking into a table or bar graph format for this section to provide more quantitative comparisons.**

We regret any misunderstanding; the statement or caveat here refers to the challenge of comparing the response only in the tropical ocean given that ENSO cycles have a strong influence on ocean productivity. We have now reordered the text in this section to make it clearer for the reader and provided more insights into the mechanisms driving the differences between ACCESS-ESM1, observations and CMIP5. We are keen to stick with the plots to highlight the differences in both the magnitude and phase of the seasonal cycle. Additionally, we have also added that the plots and text refer to the observational period 1998-2005, consistent with Anav et al (2013). The manuscript now states:

*To assess the seasonal anomaly of Net Primary Production (NPP), calculated as the anomaly of vertically integrated primary productivity through the water column, the global ocean is broken down into 5 regions, following Anav et al (2013). Figure 12 shows the NPP seasonal anomaly from ACCESS-ESM1, CMIP5 models and SeaWIFS over the (SeaWIFS) observational period 1998-2005. At the global ocean scale, seasonally we see that the magnitude of NPP from ACCESS-ESM1 is less than the amplitude of CMIP5 and SeaWIFS, with poor phasing. This likely reflects the biases in ACCESS-ESM1 toward lower latitudes, reflecting excess nutrient supply, and utilization, to the upper oligotrophic ocean (Law et al 2015) associated with deeper than observed mixed layers.*

*In the northern and southern subtropical gyres ACCESS-ESM1 (18N-49N and 19S-44S respectively) appears to overestimate the amplitude of the observed seasonal cycle when compared with SeaWIFS. Again this overestimate of NPP is associated with deeper than observed mixed layers which increase nutrient supply to the oligotrophic upper ocean. The phase of the NPP in these regions, where agreement between observations and CMIP5 is very good, is delayed by about three months. This delay may also be explained by a combination of higher (than observed) concentrations of nutrients and slower than expected biological productions associated with cool biases, particularly in the Atlantic Ocean allowing the bloom to occur later.*

*In the high latitude northern hemisphere, the magnitude of the seasonal cycle of NPP is not well captured in ACCESS-ESM1. While CMIP5 appears also to underestimate the magnitude of the seasonal cycle, ACCESS-ESM1 is lower again. In contrast in the Southern Ocean the amplitude of the seasonal cycle of NPP in ACCESS-ESM1 shows good agreement with observations. However in the high latitude oceans the phase of NPP is delayed by about 2 months. This delay may be attributed to the too shallow mixed layers that exist in these regions, which means that it is only when mixed layers start to deepen that biological productivity can start to occur. As a result the remaining growing season is shorter (than observed) leading to a reduced total productivity. This may in part explain why the total NPP northern hemisphere is much less than observed.*

*Interestingly, in the tropical ocean we see very good agreement in the amplitude of the seasonal cycle with CMIP5 and SeaWIFS. We note however, that comparing the phase of the seasonal cycle from ESMs (ACCESS-ESM1 and CMIP5) with SeaWIFS is not very meaningful in this region as they all simulate their own ENSO cycle with their own timing. Therefore any comparison over a 20 year period between models has the potential to be biased by the number of El Nino or La Nina events.*

**I also want to note on Figure 13 the Southern Ocean has different values on the x- axis, which was not mentioned. This was confusing because at first it appears the Southern Ocean does not have a seasonal cycle, but from what I can tell (which is difficult given the large range), the amplitude is comparable to the other oceanic regions.**

Clearly it is desirable to have all of the plots on the same scale. However, in order to highlight all features, different scales cannot be avoided. We have now highlighted in the captions that we use different scales in Fig.13.

Ocean Sea-Air CO2 Fluxes
**In this discussion, the authors state that the Southern subtropical gyres overestimate the observed sea- air flux; however, when I look at the figure, it looks to me the biggest uptake is occurring on the coastlines. I personally would like more of a mention of this, and maybe a discussion on why coastlines, particularly off the west coast of Australia, are taking up so much carbon compared to the broader ocean gyres.**

I think that role of the coastal ocean in the global carbon cycle is very interesting and has been suggested by authors such as Borges et al (2005). However coastal oceans in ESMs is not well represented in terms of the key mechanisms in CMIP5 models (Bopp et al, 2013). Nevertheless, despite the claims of some authors e.g. Borges et al (2005) that the coastal oceans are playing a very large role, the role likely remains small relative to the gyres, primarily because the surface area of the coastal ocean remains very small relative to the size of the ocean gyres. Furthermore, the heterogeneous response of the coastal ocean means that observationally the response coastal ocean remains more poorly constrained than the ocean. Nevertheless, as more observations are collected, resolution improves in ESMs allowing a better the representation of the key processes in the coastal ocean well better it is likely these models will focus more on these regions, as this is where the impacts of climate change and variability are felt most acutely.

Borges, A. V., Bruno Delille, and M. Frankignoulle. "Budgeting sinks and sources of CO2 in the coastal ocean: Diversity of ecosystems counts." *Geophysical Research Letters* 32.14 (2005).

**In the second paragraph of this section, the second sentence beginning with "Furthermore it appears that globally…." doesn't make sense to me, as I don't understand what it is that lies outside the range. Usually regional analyses reveal why global results occur, and it looks like the Southern Hemisphere is the main contributor to the ACCESS- ESM1 global seasonal flux anomaly? The text does go on to state this, so you may just consider removing or modifying that sentence. Later in that same paragraph, the text states that the Northern Hemisphere has fluxes larger than observed but within the range of CMIP5; however, to me it looks like there are several months when ACCESS- ESM1 is outside the range (i.e. Jan, Feb, May). Since the same disclaimer was put on this paragraph (which again I would suggest moving to the beginning of the paragraph if these figures are kept), I would consider a different format, such as a table or bar graph, or even keeping the seasonal figures but not showing the seasonal cycle in the observations if you don't believe them, and instead using a solid line.**

We have rewritten this section to clarify our intended meaning, we have also used the regional plots to explain where and why the spatial biases exist. As above we have also chosen to keep the figures as these are both helpful in our explanation and consistent with previous CMIP5 assessment paper e.g. Anav et 2013. The text now states:

*The anomaly of the seasonal cycle of the sea-air $CO_2$ fluxes was assessed against observations of W13 and CMIP5, shown in Fig 14 for the period 1986-2005. Here we see that ACCESS-ESM1 has larger global amplitude of sea-air CO2 fluxes than observed (W13) and simulated but close to the upper value of the range from CMIP5 models. We also see that globally the phase of sea-air CO2 fluxes is not well captured in ACCESS-ESM1, lying outside the range of the CMIP5 models. To better understand why there are differences between ACCESS-ESM1, CMIP5 and W13 we separate the response of sea-air CO2 into the same regions as for NPP, again following Anav et al (2013).*

*ACCESS-ESM1 appears to capture well the phase of sea-air CO2 fluxes in the subtropical gyres. In the northern subtropical gyre in particular, we see that the amplitude and phase of the seasonal cycle in ACCESS-ESM1 shows very good agreement with W13, in contrast with other ESMs (CMIP5). In the southern subtropical gyres, while the ACCESS-ESM1 appears to overestimate the amplitude relative to the observations, we see very good agreement with CMIP5 models. As anticipated the tropical ocean shows very little seasonality, nevertheless we do see good agreement with CMIP5 models. However, the comparison of ACCESS-ESM1 against observations (while shown) is not very meaningful as W13 is based on values of oceanic pCO2 from Takahashi et al, 2009 which does not include El Nino years.*

*The largest differences are seen in the representation of sea-air CO2 fluxes in the high latitude ocean. In the high latitude northern hemisphere, we see that the magnitude is larger than either CMIP5 or W13 and shows poor phasing. While the magnitude of the seasonal cycle in the Southern Ocean lies within the upper range of CMIP5 again poor phasing is seen. That the seasonal cycle is out of phase suggests that during the summer the solubility response likely dominates over the NPP response, leading to an out-gassing in the summer and uptake in the winter, as discussed in Lenton et al (2013). Consequently, we see that the poor global phasing in global sea-air CO2 fluxes is likely due to the solubility dominated response of the high latitudes during the summer.*

**Anthropogenic Inventory**
**The discussion is again quite subjective. Also, in Figure 15, there is a reference to Key et al. (2004) that is confusing and not in the text (I'm unsure if the figure appears in both papers?) Upon finishing the paper, I see the reference in the conclusions, but you might want to somehow clarify or include it in the text during this discussion.**

Key et al (2004) refers to the data used in the paper to compare with ACCESS-ESM1 while the estimated anthropogenic $CO_2$ uptake is from Sabine et al (2004). We have ensured that the correct and appropriate references are used in the figure caption, section 5.2.3 and the conclusions.

**Atmospheric CO2**
**I think it would be better to combine the first paragraph with the next sentence following it**
**(starting "Therefore, our…").**

We will do this in a revised version.

**Conclusions**
**I think the conclusions would be stronger if some quantitative assessments were provided. I also noticed the order of the conclusions was not the same as the order presented in the text, but neither was it combined in a more succinct fashion to group conclusions. Simply reordering the paragraphs would make the text more consistent.**

We will reorder the paragraphs in a revised version and also include some quantitative assessments.

**In the second- to- last paragraph, I was confused by the statement that "Seasonally the ACCESS- ESM1 appears biased toward the Southern Hemisphere." I'm not sure how to interpret this statement, as to me it's using both temporal and spatial references combined in an unclear fashion. This same type of statement occurs later in the sentence "Globally the annual mean is well captured but biased to low latitudes." I recommend reworking those sentences to clarify the intent of the text. Also, a quantification of how it is "well captured" would strengthen the statement.**

We have now rephrased this paragraph to better reflect our intended meaning and removed the very qualitative comments. We have not added numbers to the conclusions but instead have added more quantitative analysis to the text that is referred to in the conclusion, the text now states:

*Globally integrated sea-air CO2 fluxes are well captured and we reproduce very well the cumulative uptake estimate from the Global Carbon Project (LeQuere et al, 2014) and our anthropogenic uptake agrees very well with observed GLODAP value of  Sabine et al (2004). The spatial distribution of sea-air CO2 fluxes is also well reproduced by CMIP5 models and observations.  At the same time global ocean NPP also shows good agreement with observations and lies well within the range of CMIP5 models. However seasonal biases do exist in sea-air CO2 fluxes and NPP, potentially related to biases in mixed layer depth and surface temperature that are present in ACCESS-ESM1; and will need to be addressed in later versions of ACCESS-ESM1.*

**Technical Comments**
**Overall I have very few technical comments. My main technical comments are on comma usage. I personally find well- placed commas can aid in reading the text, but am not a comma expert, so feel free to take or leave the suggestions.**

All technical comments will be considered and corrected in a revised version.

---

## Author Response (AR2)

Response to handling editor and reviewer 4 (part 1):

Most of the comments raised by the editor and reviewer 4  concern part 1 of the manuscript. A comprehensive reply is therefore provided with the submission of a revised version for part 1.

Reviewer 4 asked for a clear focus on evaluation for the ocean vs. sensitivity tests. Following consultation with the editor, we have therefore moved the evaluation of ocean surface fields against observations and CMIP5 models (including Figures 12 and 13) to part 2.

[revised manuscript text omitted]
$ | 1762 | 88.6 | 10.0 | 2067 | 93.3 | 12.3 | 1846 | 90.5 | 10.5 | 2187 | 96.4 | 13.0 | 97.6 | 136.5 |